**1** Insights into particulate matter pollution in the North China Plain during wintertime:
**2** Local contribution or regional transport?

**4** Jiarui Wu[1,4], Naifang Bei[2], Yuan Wang[3], Xia Li[1,4], Suixin Liu[1,4], Lang Liu[1,4], Ruonan Wang[1,4], Jiaoyang
**5** Yu[1], Tianhao Le[3], Min Zuo[1,4], Zhenxing Shen[2], Junji Cao[1,4], Xuexi Tie[1], and Guohui Li[1,4*]

**7** [1]Key Lab of Aerosol Chemistry and Physics, SKLLQG, Institute of Earth Environment, Chinese Academy
**8** of Sciences, Xi'an, 710061, China
**9** [2]School of Human Settlements and Civil Engineering, Xi'an Jiaotong University, Xi'an, 710049, China
**10** [3]Division of Geological and Planetary Sciences, California Institute of Technology, Pasadena, CA 91125,
**11** USA
**12** [4]CAS Center for Excellence in Quaternary Science and Global Change, Xi'an, 710061, China

**14** *Correspondence to: Guohui Li (ligh@ieecas.cn)

**17** **Abstract.** Accurate identification and quantitative source apportionment of fine particulate

**18** matters ($PM_{2.5}$) provide an important prerequisite for design and implementation of emission

**19** control strategies to reduce PM pollution. Therefore, a source-oriented version of the

**20** WRF-Chem model is developed in the study to make source apportionment of $PM_{2.5}$ in the

**21** North China Plain (NCP). A persistent and heavy haze event that occurred in the NCP from

**22** 05 December 2015 to 04 January 2016 is simulated using the model as a case study to

**23** quantify $PM_{2.5}$ contributions of local emissions and regional transport. Results show that

**24** local and non-local emissions contribute 36.3% and 63.7% of the $PM_{2.5}$ mass in Beijing

**25** during the haze event on average. When Beijing's air quality is excellent or good in terms of

**26** hourly $PM_{2.5}$ concentrations, local emissions dominate the $PM_{2.5}$ mass with contributions

**27** exceeding 50%. However, when the air quality is severely polluted, the $PM_{2.5}$ contribution of

**28** non-local emissions is around 75%. The non-local emissions also dominate the Tianjin's air

**29** quality, with average $PM_{2.5}$ contributions exceeding 65%. The $PM_{2.5}$ level in Hebei and

**30** Shandong is generally controlled by local emissions, but in Henan, local and non-local

**31** emissions play an almost equivalent role in the $PM_{2.5}$ level, except when the air quality is

**32** severely polluted, with non-local $PM_{2.5}$ contributions of over 60%. Additionally, the primary

**33** aerosol species are generally dominated by local emissions with the average contribution

**34** exceeding 50%. However, the source apportionment of secondary aerosols shows more

**35** evident regional characteristics. Therefore, except for cooperation with neighboring

**36** provinces to carry out strict emission mitigation measures, reducing primary aerosols

**37** constitutes the priority to alleviate PM pollution in the NCP, especially in Beijing and

**38** Tianjin.

## 1 Introduction

As the most polluted area in China, the North China Plain (NCP) has been suffering from severe particulate pollution for recent decades, particularly during wintertime, caused by a synergy of local emissions, trans-boundary transport, specific topography, and unfavorable meteorological situations (Long et al., 2016; Wu et al., 2017; An et al., 2019; Wu et al., 2020). In recent years, the Chinese government has carried out aggressive emission mitigation measures to reduce particulate matter (PM) pollution (Zheng et al., 2018; Zhang et al., 2019), but heavy haze with high $PM_{2.5}$ (fine PM) concentrations still frequently engulfs the area. It is controversial on whether local emissions or trans-boundary transport dominates the PM pollution in the NCP, especially in Beijing (Guo et al., 2014; Li et al., 2015; Zhang et al., 2015; Wu et al., 2017; Zamora et al., 2019). Therefore, accurate identification and quantitative source apportionment (SA) of $PM_{2.5}$ is imperative to provide scientific reference for instituting air quality control strategies as well as constitute an important prerequisite to reduce PM pollution in the NCP.

The observation based SA techniques, such as chemical mass balance (CMB) and positive matrix factorization (PMF) methods, are traditionally used to quantify the particle contribution of each source (Cooper and Watson, 1980; Paatero and Tapper, 1993), but they cannot identify the source contribution of secondary transformation to particulate matters. The brute force method (BFM) is the simplest model based SA method using air quality models (AQMs) through zeroing out emissions from a specific source (Marmur et al., 2005). The BFM can assess the importance of each emission source, but has flaws in quantifying the source contribution due to lack of consideration of the complicated non-linear interaction between various sources (Zhang and Ying, 2011). At present, the widely used SA technique based on AQMs is the reactive tracer method or the source-oriented AQMs (Marmur et al., 2006; Ying and Kleeman, 2006; Ying et al., 2008a; Ying et al., 2008b; Zhang and Ying, 2010,

**66** 2011; Burr and Zhang, 2011; Zhang et al., 2014). The method adds reactive tracers or tagged

**67** species in AQMs to trace the atmospheric transport, transformation, and deposition of air

**68** pollutants emitted from specific sources and quantify the source contribution according to the

**69** mass conservation (Wagstrom et al., 2008; Wang et al., 2009).

**70** The observation based SA method or the BFM based on AQMs has been used to

**71** evaluate $PM_{2.5}$ contributions of local emissions and trans-boundary transport in the NCP,

**72** especially in Beijing-Tianjin-Hebei (BTH). Chang et al. (2019) have investigated the

**73** contribution of trans-boundary transport to the $PM_{2.5}$ concentration in 13 cities of the BTH,

**74** showing that Shandong province has a considerable $PM_{2.5}$ contribution to most cities in BTH,

**75** followed by Henan among the four neighboring provinces. Dong et al. (2020) have also

**76** found that the regional transport contributes about 32.5%-68.4% of $PM_{2.5}$ concentrations in

**77** BTH in 2017. However, the contribution of local emissions or trans-boundary transport to

**78** Beijing's PM pollution still remains uncertain. Lang et al. (2013) have indicated that regional

**79** transport accounts for 54.6% of $PM_{2.5}$ concentrations during polluted episodes in Beijing,

**80** with annual $PM_{2.5}$ contribution of 42.4% on average using the observation and MM5–CMAQ

**81** model results. Wu et al. (2017) have shown that non-Beijing emissions contribute 61.5% of

**82** $PM_{2.5}$ mass during haze events in summer. However, some studies have emphasized that

**83** severe haze formation that occurs in Beijing is mainly controlled by the efficient local aerosol

**84** nucleation and growth, whereas the $PM_{2.5}$ contribution of regional transport might not be

**85** significant (Guo et al., 2014; Zamora et al., 2019). Meng et al. (2020) found that the regional

**86** transport from Hebei and Shandong plays an important role in the PM pollution in Tianjin,

**87** with the average $PM_{2.5}$ contribution of 44% during the wintertime, but the local contribution

**88** gradually dominates with continuous deterioration of the PM pollution. Wang et al. (2015)

**89** have concluded that regional transport plays a non-negligible role in the top three polluted

**90** cities in Hebei using the BFM method, with $PM_{2.5}$ contributions of 27.9% in Shijiazhuang,

46.6% in Xingtai, and 40.4% in Handan. However, Wang et al. (2019) have proposed that local emissions are the main contributor to the air pollution in Hebei. Liu et al. (2017) have emphasized that the contribution of regional transport to the PM pollution in Henan is significant during the wintertime, with the average $PM_{2.5}$ contribution of 11.95%, 11.69%, 7.95%, and 7.4% from BTH, Anhui, Jiangsu, and Shandong, respectively. In summary, these studies suggest that there is uncertainty regarding whether local contribution or regional transport is dominant during PM pollution events in the NCP.

In this study, a source-oriented WRF-Chem model is developed to comprehensively quantify the contribution of local emissions and trans-boundary transport to the PM pollution in the NCP, including Beijing, Tianjin, Hebei, Henan, and Shandong, as well as the adjacent province on the west, Shanxi, under different pollution levels during the wintertime in 2015. The model and methodology are described in Section 2. The results and discussions are presented in Section 3, and summary and conclusions are given in Section 4.

## 2    Model and methodology

### 2.1  WRF-Chem model and configurations

The source-oriented AQM used in this study is based on the WRF-Chem model (Version 3.5) (Grell et al., 2005) with modifications by Li et al. (2010, 2011a, 2011b). The modified WRF-Chem model includes a new flexible gas phase chemical module that can be used with different chemical mechanisms and the CMAQ aerosol module (AERO5) developed by the US EPA (Binkowski and Roselle, 2003; Foley et al., 2010). The wet deposition is based on the method in the CMAQ module and the dry deposition of chemical species follows Wesely (1989). The photolysis rates are calculated using the Fast Tropospheric Ultraviolet and Visible (FTUV) Radiation Model with the aerosol and cloud effects on photolysis (Li et al., 2005; Li et al., 2011a). The inorganic aerosols are predicted

**116** using ISORROPIA Version 1.7, calculating the composition and phase state of an

**117** ammonium-sulfate-nitrate-water inorganic aerosol in thermodynamic equilibrium with gas

**118** phase precursors (Nenes et al., 1998). The secondary organic aerosols (SOA) are calculated

**119** using the volatility basis-set (VBS) modeling method, with contributions from glyoxal and

**120** methylglyoxal. Detailed information can be found in Li et al. (2010, 2011a, 2011b). Figure 1

**121** shows the simulation domain and detailed model configuration can be found in Table 1. It is

**122** worth noting that the emission inventory used in this study is developed by Zhang et al. (2009)

**123** and Li et al. (2017) with the base year of 2012. Considering that the great changes in

**124** emission inventory due to implementation of the toughest-ever clean air policy in China

**125** (Zhang et al., 2019), the emission inventory has been adjusted according to the trends from

**126** 2012 to 2015 proposed by Zheng et al. (2018).

**127** **2.2 Source-Oriented WRF-Chem model**

**128** In the source-oriented WRF-Chem model, the SAPRC-99 photochemistry mechanism

**129** (Carter, 2010) and CMAQ aerosol module (AERO5) (Foley et al., 2010) are modified so that

**130** the precursors of aerosols from different sources and their corresponding reaction products

**131** are treated as different species and tracked independently in chemical, physical, and

**132** dynamical processes. It is worth noting that the tagged species have exactly identical physical

**133** and chemical properties as the original ones.

**134** Black carbon (BC) and unspecified species (mainly mineral dust) from each source are

**135** tagged and only tracked in processes of transport, dispersion, and wet/dry deposition since

**136** they do not involve in photochemistry and gas-to-particle partitioning. For the inorganic

**137** aerosols (sulfate, nitrate, and ammonium) and organic aerosols (primary and secondary

**138** organic aerosols, i.e., POA and SOA), their precursors from each source and corresponding

**139** reaction products are treated as different species and simulated in the SAPRC-99

**140** photochemistry mechanism and traced in processes of transport, dispersion, and wet/dry

deposition as well as gas-to-particle partitioning. A non-hardwired gas phase chemical
module is used to solve the SAPRC-99 photochemistry based on the Eulerian backward
Gauss-Seidel iterative technique (Hess et al., 2000; Li et al., 2010). The module is flexible to
include a new gas-phase species and its corresponding photochemical reactions.
The ISORROPIA is used to distribute the $NH_3$/ammonium, $HNO_3$/nitrate, and water
between the gas and aerosol phases as a function of total sulfate, total ammonia, total nitrate,
relative humidity and temperature (Nenes et al., 1998). Therefore, as a bulk method, the
ISORROPIA cannot be applied to distribute the gas and aerosol phase for the inorganic
aerosol from each source separately because of the interaction among various sources.
Except primary emissions, the SA for sulfate aerosols needs to be considered in the
homogeneous and heterogeneous formation pathways. The sulfate growth from the gas-phase
$SO_2$ oxidation is contributed by the $H_2SO_4$ involved nucleation and condensation, which are
determined by the $H_2SO_4$ formation rate in the atmosphere. At time ($t$), after one time step
($\delta t$) integeration, the conceptual scheme of the source-oriented sulfate gas-phase formation is
shown in Figure 2a. In this study, a $SO_2$ heterogeneous reaction parameterization associated
with aerosol water is used, in which the $SO_2$ oxidation in aerosol water by $O_2$ catalyzed by
$Fe^{3+}$ is limited by mass transfer resistances in the gas-phase and the gas-particle interface.
Considering the effect of ionic strength and aerosol water acidity, the sulfate heterogeneous
formation from $SO_2$ is therefore parameterized as a first-order irreversible uptake by aerosols,
with a reactive uptake coefficient of $0.5\times10^{-4}$, assuming that there is enough alkalinity to
maintain the high iron-catalyzed reaction rate (Li et al., 2017). The detailed description of the
parameterization of the heterogeneous oxidation of $SO_2$ involving aerosol water can be seen
in Supplement. Figure 2b presents the sulfate SA for the heterogeneous formation. It is worth
noting that, although it is lack of precipitation during the simulated episode, the SA of sulfate
formed in cloud water is also considered. The $SO_2$ in cloud water is oxidized mainly by $H_2O_2$,

**166** $O_3$, $NO_2$, formic acid, and $O_2$ catalyzed by $Fe^{3+}$ and $Mn^{2+}$. The SA for nitrate and ammonium

**167** aerosols follows the mass conversion of $N(+VI)$ and $N(-III)$ from each source,

**168** respectively, when the total ammonia and nitrate are distributed between the gas and aerosol

**169** phases by the ISORROPIA after one time step integration, as shown in Figure 3.

**170** Organic aerosols are simulated using a non-traditional SOA module based on the

**171** volatility basis set (VBS) method, in which all primary species are treated as chemically

**172** reactive and distributed in logarithmically spaced volatility bins (Donahue et al., 2006;

**173** Robinson et al., 2007). Nine surrogate species with saturation concentration ranging from

**174** $10^{-2}$ to $10^6$ μg m$^{-3}$ at room temperature are considered to represent POA compositions

**175** (Shrivastava et al., 2008). The SOA formation from anthropogenic or biogenic precursors is

**176** predicted using four semi-volatile organic compounds (SVOCs) whose effective saturation

**177** concentrations at room temperature are 1, 10, 100, and 1000 μg m$^{-3}$, respectively (Tsimpidi et

**178** al., 2010). The SOA formation includes the following pathways: (1) the oxidation of VOCs

**179** emitted from anthropogenic and biogenic sources, (2) intermediate VOCs (IVOCs)

**180** co-emitted with POA but are never in the particle phase during the emissions process

**181** oxidized by OH, and (3) primary organic gases (POG) emitted or formed due to evaporation

**182** of POA assumed to react with OH radicals to reduce their volatility and hence to partition

**183** between gas and particle phase forming SOA (Odum et al., 1996; Pankow, 1994; Lipsky and

**184** Robinson, 2006; Robinson et al., 2007; Shrivastava et al., 2006). The SOA yield from VOC$_s$

**185** is NO$_x$ dependent (Li et al., 2011a). The high-NO$_x$ and low-NO$_x$ yields are listed in the Table

**186** S1 and parameters used to treat partitioning of POA emissions are listed in Table S2. The

**187** VBS method is in principle source-oriented, which can be used to trace the OA formation

**188** from various sources. Therefore, when considering SA for organic aerosols, we just need to

**189** treat all the SOA and POA, as well as their corresponding gas-phase organics from each

**190** emission source, as the VBS input, as shown in Figure 4a. For the heterogeneous pathway,

**191** the SOA formation from glyoxal and methyglyoxal is parameterized as a first-order

**192** irreversible uptake on aerosol or cloud droplet surfaces with a reactive uptake coefficient of

**193** $3.7 \times 10^{-3}$ (Volkamer et al., 2007; Zhao et al., 2006). The SA for heterogeneous SOA

**194** formation is shown in Figure 4b, which is similar to that for heterogeneous sulfate formation.

**195** **2.3  Data and statistical methods for comparisons**

**196** The model performance in simulating $PM_{2.5}$, $O_3$, $NO_2$, $SO_2$, and CO is validated using

**197** the hourly observations released by Ministry of Ecology and Environment of China (China

**198** MEP), with 389 observation sites in the NCP. In addition, the predicted submicron sulfate,

**199** nitrate, ammonium, and organic aerosols are compared to measurements by the Aerodyne

**200** Aerosol Chemical Speciation Monitor (ACSM), which is deployed at the National Center for

**201** Nanoscience and Technology (NCNST), Chinese Academy of Sciences in Beijing (Figure 1).

**202** POA and SOA concentrations are obtained from the ACSM measurement analyzed using the

**203** PMF. The meteorological parameters including surface pressure, temperature, wind speed

**204** and    direction    with    a    3-hour    interval    are    obtained    from    the    website

**205** http://www.meteomanz.com, including the observation sites at Beijing, Tianjin, Shijiazhuang,

**206** Jinan, Zhengzhou, Hefei, and Nanjing (Figure S1). Furthermore, the reanalysis data from the

**207** European Centre for Medium-Range Weather Forecasts (ECMWF) are used to analyze the

**208** synoptic patterns during the study episode.

**209** In the present study, the mean bias (*MB*), root mean square error (*RMSE*) and the index

**210** of agreement (*IOA*) are used as indicators to evaluate the performance of the WRF-Chem

**211** model. *IOA* describes the relative difference between the model and observation, ranging

**212** from 0 to 1, with 1 indicating perfect agreement.

**213** $$MB = \frac{1}{N}\sum_{i=1}^{N}(P_i - O_i)$$

**214** $$RMSE = \left[\frac{1}{N}\sum_{i=1}^{N}(P_i - O_i)^2\right]^{\frac{1}{2}}$$

**215**
$$IOA = 1 - \frac{\sum_{i=1}^{N}(P_i - O_i)^2}{\sum_{i=1}^{N}(|P_i - \overline{O}| + |O_i - \overline{O}|)^2}$$

**216** Where $P_i$ and $O_i$ are the predicted and observed pollutant concentrations, respectively. $N$ is

**217** the total number of the predictions used for comparisons, and $\overline{P}$ and $\overline{O}$ represents the

**218** average of the prediction and observation, respectively.

**219**

**220** ## 3    Results and discussions

**221** ### 3.1  Model performance

**222**    Figure 5 shows the diurnal profiles of observed and simulated near-surface $PM_{2.5}$, $O_3$,

**223** $NO_2$, $SO_2$ and CO concentrations averaged at monitoring sites in the NCP from 05 December

**224** 2015 to 04 January 2016. The model generally performs well in reproducing the temporal

**225** variation of $PM_{2.5}$ concentrations in the NCP, with an *IOA* of 0.96, but slightly overestimates

**226** $PM_{2.5}$ concentrations against measurements, with a *MB* of 2.2 μg m$^{-3}$. The diurnal $O_3$

**227** variation is successfully replicated by the model, such as peak afternoon $O_3$ concentrations

**228** caused by active photochemistry and low nighttime $O_3$ concentrations due to the $NO_x$ titration,

**229** with an *IOA* of 0.88. However, the model is subject to underestimating the $O_3$ concentration

**230** compared to measurements, particularly during nighttime, with a *MB* of -5.9 μg m$^{-3}$. The

**231** model also reasonably well reproduces the $NO_2$ diurnal profiles with peaks in the evening,

**232** with an *IOA* of 0.89 and a *MB* of 0.5 μg m$^{-3}$, but considerable overestimations or

**233** underestimations still exist. The model generally tracks reasonably the temporal variation of

**234** $SO_2$ concentrations against observations, with an *IOA* of 0.76. However, the biases for the

**235** $SO_2$ simulation are also large considering that $SO_2$ is mainly emitted from point sources and

**236** its simulations are more sensitive to the wind field uncertainties (Bei et al., 2017). Compared

**237** with measurements, the temporal profile of the near-surface CO concentration in the NCP is

**238** well simulated, with the *IOA* and *MB* of 0.90 and 0.0 μg m$^{-3}$, respectively. Generally, the

**239** accumulation and trans-boundary transport of air pollutants is mainly dependent on regional

**240** meteorological conditions. Figure S2 shows the average geopotential heights at 500hPa and

**241** the mean sea level pressures with wind vectors during the study episode. During the

**242** simulated episode, the NCP is situated behind the trough at 500 hPa. The NCP is controlled

**243** by the high pressure system at the surface on a large scale due to the upper level trough,

**244** ranging from 1026 to 1030 hPa, and the prevailing wind over the NCP is weak or calm,

**245** which is unfavorable for dissipation of air pollutants. Figure S3 shows the diurnal profiles of

**246** observed and simulated near-surface pressure, temperature, wind speed, and wind direction

**247** averaged at monitoring sites in the NCP from 05 December 2015 to 04 January 2016. The

**248** WRF-Chem model performs well in reproducing the diurnal variability of near surface

**249** pressure, surface temperature (TSFC), wind speed, and wind direction, with *IOA*s of 0.63,

**250** 0.84, 0.75, and 0.54, respectively. During the study episode, the simulated and observed of

**251** near surface pressures are 1024.0hPa and 1028.5hPa, indicating that a high pressure system

**252** controlling the NCP (Figure S2). The southerly wind prevails over the NCP during the study

**253** episode, with the simulated and observed wind direction of 180.6° and 175.1°. Moreover, the

**254** simulated and observed wind speed is approximately 2 m s$^{-1}$ over the NCP during the

**255** simulated episode. Therefore, the air pollutants are subject to being transported from south to

**256** north, and the weak or calm wind also appears in some regions, which is favorable for the

**257** accumulation of air pollutants. For example, from 16 to 24 December 2015, the wind speed

**258** in the NCP decreases and the wind direction turns to be southerly, facilitating accumulation

**259** of air pollutants, and meanwhile a serious PM pollution episode with high $PM_{2.5}$

**260** concentrations occurs.

**261** Figure 6 shows the spatial pattern of simulated and observed average near-surface

**262** concentrations of $PM_{2.5}$, $O_3$, $NO_2$, and $SO_2$ along with simulated winds during the episode in

**263** the NCP. The simulated air pollutants distributions are generally in good agreement with

**264** observations, although the model biases still exist. During the haze episode, the simulated

weak or calm winds are favorable for accumulation of air pollutants, leading to formation of
the serious air pollution in the NCP. The simulated average near-surface $PM_{2.5}$ concentrations
during the episode are more than 115 μg m$^{-3}$ in the NCP, which is consistent with
observations. The simulated and observed average $O_3$ concentrations during the episode are
not high, generally less than 40 μg m$^{-3}$. The low $O_3$ concentration during the episode is
chiefly caused by the slow photochemical activities due to weak wintertime insolation which
is further attenuated by clouds and aerosols and the resultant titration of high $NO_x$ emissions
(Li et al., 2018). The observed and calculated average $NO_2$ and $SO_2$ concentrations are still
high in the NCP, varying from 30 to 100 μg m$^{-3}$ and 20 to 100 μg m$^{-3}$, respectively, although
strict emission mitigation measures have been carried out since 2013. Interestingly, the
simulated $SO_2$ concentrations in cities and their surrounding areas are very high, but the
simulated $NO_2$ concentrations present uniform distribution in the NCP, indicating the
substantial contribution of $NO_x$ area sources. The diurnal variability in the spatial distribution
of simulated and observed air pollutants is shown in Figures S9 to S12. The spatial patterns
of air pollutants at different time are generally similar to those of the episode average. The
$PM_{2.5}$ pollution in the NCP is more severe during nighttime and early morning, especially at
08:00 and 20:00 BJT due to the rush hour.

282       Figure 7 provides the temporal variations of simulated and observed aerosol species at

NCNST in Beijing during the episode. Generally, the model predicts reasonably the temporal
variations of the aerosol species against the measurements. The model yields the major peaks
of the POA concentration compared to observations in Beijing, but frequently underestimates
or overestimates the POA concentration, with an *IOA* of 0.80 and a *MB* of -2.0 μg m$^{-3}$. As a
primary species, the POA in Beijing is determined by local emissions and regional transport
outside of Beijing during haze days, so uncertainties from emissions and meteorological
fields have large potential to influence POA simulations (Bei et al., 2017). Although the VBS

**290** modeling method is used and contributions from glyoxal and methylglyoxal are included in

**291** the study, the model still has difficulties in simulating the SOA concentrations, with the *IOA*

**292** and *MB* of 0.67 and -10.5 $\mu g\ m^{-3}$, respectively. Except the SOA formation and transformation

**293** mechanism in the atmosphere, which remains elusive, many factors have the potential to

**294** affect the SOA simulation, such as meteorology, measurements, precursor emissions, and

**295** SOA treatments (Li et al., 2011). The model reasonably reproduces the sulfate temporal

**296** variation compared to measurements, and the *MB* and *IOA* are -3.5 $\mu g\ m^{-3}$ and 0.87,

**297** respectively. The model also performs well in simulating the nitrate and ammonium

**298** concentrations against measurements in Beijing, with *IOAs* of 0.92 and 0.88, respectively.

**299** Generally, the model simulates well the spatial distribution and temporal variation of air

**300** pollutants in the NCP, and the predicted aerosol species are also consistent with the

**301** measurements in Beijing. Good model performance in simulating air pollutants and aerosol

**302** species provides a reliable base for quantifying contributions of local and non-local emissions

**303** to the PM pollution in the NCP.

**304** **3.2 Source apportionment of the PM pollution in the NCP**

**305** We have marked the emitted precursors in six provinces, including Beijing, Tianjin,

**306** Hebei, Henan, Shandong, and Shanxi in simulations of the source-oriented WRF-Chem

**307** model (Figure S1). Additionally, the boundary transport and emissions from areas not within

**308** the six provinces are taken as the background source. Therefore, $PM_{2.5}$ contributions of the

**309** non-local emission for each of the six provinces include those transported from the other five

**310** provinces and the background source.

**311** Figure 8 shows the average $PM_{2.5}$ contribution of emissions from the six provinces

**312** during the study episode. Apparently, emissions from the six provinces influence the $PM_{2.5}$

**313** level in the whole NCP, showing necessity of collaborative emission mitigation to reduce PM

**314** pollution. Emissions of Hebei, Henan, and Shandong not only significantly deteriorate the

local PM pollution, with $PM_{2.5}$ contributions ranging from 50 to over 100 μg m$^{-3}$, but also
considerably enhance the $PM_{2.5}$ level in their surrounding areas by about 5~50 μg m$^{-3}$.
Emissions of Beijing and Tianjin increase the local $PM_{2.5}$ concentrations by 10~100 μg m$^{-3}$,
and contribute about 3~10 μg m$^{-3}$ $PM_{2.5}$ to their surrounding areas. Due to blocking of
mountains, $PM_{2.5}$ contributions of the Shanxi emission to the NCP is not significant, ranging
from 3 to 20 μg m$^{-3}$. The diurnal variations in the spatial distribution of average $PM_{2.5}$
contributions from the six provinces during the study episode are also shown in Figures S14
to S19. There is no significant difference among the spatial distribution of $PM_{2.5}$ contributions
from the six provinces at different time, but the higher $PM_{2.5}$ contribution of emissions from
the source region generally occurs at 08:00 and 20:00 BJT.
Beijing is surrounded from the southwest to the northeast by the Taihang Mountains and
the Yanshan Mountains and open to the NCP in the south and east. During haze events,
southerly or easterly winds are generally prevailed in the NCP (Figure S3), facilitating
transport of air pollutants emitted from the NCP to Beijing and further accumulation due to
the mountain blocking (Long et al., 2016). During the study episode, the average simulated
$PM_{2.5}$ concentration in Beijing is around 125.3 μg m$^{-3}$, in which the contribution of local
emissions is 36.3%. The remaining 63.7% of $PM_{2.5}$ concentrations in Beijing is accounted for
by non-Beijing emissions, showing that Beijing's air quality is dominated by non-Beijing
emissions during the PM pollution episode. The $PM_{2.5}$ contribution of Hebei emissions to
Beijing is 24.6%, greater than those of Shandong (8.3%), Tianjin (7.4%), Henan (3.6%), and
Shanxi (3.3%). The background source contributes about 16.5% of the $PM_{2.5}$ mass in Beijing
on average. Overall, the contribution of emissions from Beijing's five surrounding provinces
to the $PM_{2.5}$ mass is 47.2%, exceeding that of local emissions, indicating the importance of
the trans-boundary transport of air pollutants in the haze formation in Beijing. Adjacent to
Beijing, the Tianjin's air quality is also dominated by trans-boundary transport of air

pollutants. The average $PM_{2.5}$ contribution of non-local emissions is 67.3%, in which Hebei, Shandong, Beijing, Henan, and Shanxi accounts for 29.3%, 11.7%, 8.0%, 4.0%, and 3.0%, respectively. The $PM_{2.5}$ contribution of local emissions in Hebei, Henan, and Shanxi is almost as much as that of trans-boundary transport, with the average of 50.2%, 45.7%, and 49.2%, respectively. The Shandong emissions play an important role in the air quality in Hebei and Henan, with $PM_{2.5}$ contributions of about 15%. Moreover, the Shandong's air quality is primarily determined by local emissions, with an average $PM_{2.5}$ contribution of 64.9%. Emissions of Beijing, Tianjin, Hebei, Henan, and Shanxi contribute less than 8% of the $PM_{2.5}$ mass in Shandong. The background source makes up approximately 11.3%, 11.4%, 16.8%, 11.4%, and 21.8% of the $PM_{2.5}$ mass in Tianjin, Hebei, Henan, Shandong, and Shanxi, respectively. Figure S20 also provides the vertical profiles of the average $PM_{2.5}$ contribution from local and non-local emissions in Beijing, Tianjin, Hebei, Henan, Shandong, and Shanxi during the episode. Generally, the $PM_{2.5}$ contribution of local emissions in the six provinces in the NCP declines rapidly with altitude due to the efficient advection in the upper PBL. The local contribution decreases to less than 20% in the upper PBL in Beijing and Tianjin and is generally more than 25% in the other four provinces. In Shandong, the $PM_{2.5}$ concentration is mainly dominated by local emissions in the lower PBL, but the local contribution presents a significant decreasing trend in the upper PBL.

Previous studies have shown that there exist large uncertainties in the contribution of local emissions or trans-boundary transport to Beijing's PM pollution (Guo et al., 2010; Guo et al., 2014; Li et al., 2015; Zhang et al., 2015; Wu et al., 2017). We further evaluate the contribution of local and non-local emissions to the $PM_{2.5}$ mass in Beijing under different pollution levels, as well as in the other five provinces. The simulated hourly near-surface $PM_{2.5}$ mass concentrations in Beijing during the whole episode are first subdivided into 6 bins based on the air quality standard in China for $PM_{2.5}$, i.e., 0~35 (excellent), 35~75 (good),

75~115 (lightly polluted), 115~150 (moderately polluted), 150~250 (heavily polluted), and exceeding 250 (severely polluted) $\mu g \, m^{-3}$ (Feng et al., 2016). $PM_{2.5}$ contributions from local emissions and the other five provinces as well as background source to Beijing are assembled separately as the bin $PM_{2.5}$ concentrations following the grid cells, and an average of $PM_{2.5}$ contributions from each source in each bin is calculated. The same method is also used for the other five provinces.

Table 2, Table 3 and Figure 9 present the average percentage contribution of local and non-local emissions to the $PM_{2.5}$ concentrations in Beijing, Tianjin, Hebei, Henan, Shandong, and Shanxi during the episode under different pollution levels. The local emission dominates the $PM_{2.5}$ mass when the air quality is excellent and good in Beijing, with the average contribution of 56.8% and 55.0%, respectively. Moreover, the $PM_{2.5}$ contribution of local emissions decreases with the deterioration of the air quality in Beijing, with an average contribution of 48.7%, 40.5%, 35.4%, and 25.1%, respectively, when the air quality is slightly, moderately, heavily, and severely polluted. Therefore, non-local emissions play a dominant role in Beijing's PM pollution; particularly when the air quality is severely polluted, non-local emissions contribute around 75% of the $PM_{2.5}$ mass in Beijing. With the excellent and good air quality in Beijing, the contribution of emissions from the other five provinces is 22.4% and 29.5%, respectively, much less than those of local emissions. However, the contribution increases from 37.6% to 54.3% with deterioration of Beijing's air quality from being slightly to severely polluted. The result is consistent with that from Lang et al. (2013), reporting that regional transport accounts for 54.6% of the $PM_{2.5}$ mass in Beijing during a PM pollution episode. Additionally, Jiang et al. (2015) have concluded that the transport from the environs of Beijing contributes about 55% of the peak $PM_{2.5}$ concentration in the city during a severe PM pollution episode that occurred in December 2013. Wu et al. (2017) have also shown that 61.5% of the $PM_{2.5}$ mass in Beijing is contributed by regional transport during a

**390** summertime PM pollution episode. The contribution of Hebei emissions to the $PM_{2.5}$ mass in

**391** Beijing is the most significant, exceeding 20% when Beijing's air quality is not excellent.

**392** The contribution of emissions from Tianjin, Henan, Shandong, and Shanxi is generally less

**393** than 10% under different pollution levels. However, when Beijing's air quality is severely

**394** polluted, the contribution of Shandong emissions is also significant, attaining 16.4%. The

**395** background source contributes more than 20% of the $PM_{2.5}$ mass in Beijing when the air

**396** quality is excellent and severely polluted, and between 12.8% and 15.4% under the other

**397** pollution levels.

**398** The air quality in Tianjin is dominated by trans-boundary transport of air pollutants,

**399** with the non-local $PM_{2.5}$ contribution generally higher than 55%, especially when the air

**400** quality is severely polluted, with the non-local $PM_{2.5}$ contribution of 70%, which is higher

**401** than the average non-local contribution of 44% reported by Meng et al. (2020). The $PM_{2.5}$

**402** contribution of local emissions decreases with the deterioration of the air quality in Tianjin,

**403** with average contributions of 44.9%, 41.3%, 37.0%, and 29.6%, respectively, when the air

**404** quality is good, slightly, moderately, and heavily polluted. The Hebei emissions play a

**405** significant role in the PM pollution in Tianjin, generally contributing more than 25% of $PM_{2.5}$

**406** concentrations, except when the air quality is excellent. Meng et al. (2020) have emphasized

**407** the important contribution of Hebei emissions to $PM_{2.5}$ concentrations in Tianjin. However,

**408** Meng et al. (2020) have suggested that the $PM_{2.5}$ contribution of local emissions gradually

**409** increases with continuous deterioration of the PM pollution, which is different from that in

**410** this study. The $PM_{2.5}$ contribution of the background source is between 11.4% to 16.5%,

**411** except when the air quality is severely polluted, with the contribution less than 10%.

**412** The Hebei's air quality is obviously determined by local emissions when the air quality

**413** is excellent or good, with the average $PM_{2.5}$ contribution of 65.8% and 60.9%, respectively.

**414** Additionally, the contribution of non-local emissions to the $PM_{2.5}$ mass in Hebei is almost the

**415** same as that of local emissions, varying from 46.2% to 54.8% with $PM_{2.5}$ concentrations

**416** exceeding 75 μg m$^{-3}$. The $PM_{2.5}$ contribution of emissions from Tianjin, Henan, and Shanxi is

**417** generally less than 10% under different pollution levels. However, the Shandong emissions

**418** contribute more than 10% of the $PM_{2.5}$ mass in Hebei when the air quality becomes polluted.

**419** Obviously, with occurrence of severe PM pollution in BTH, the contribution of Shandong

**420** emissions to the $PM_{2.5}$ mass in BTH becomes considerable, which has also been suggested by

**421** Chang et al. (2019). The $PM_{2.5}$ contribution of background source to Hebei decreases with

**422** deterioration of the air quality, ranging from 8.2% to 19.2% during the episode. Overall, in

**423** Hebei, local emissions generally dominate the $PM_{2.5}$ level under different pollution level, but

**424** non-local emissions play an increasingly important role with deterioration of PM pollution,

**425** which is consistent with the findings of Wang et al. (2015) and Wang et al. (2019).

**426** The local and non-local emissions generally play an almost equivalent role in the air

**427** quality in Henan when the severe PM pollution does not occur. However, when the air quality

**428** is severely polluted, the non-local emissions contribute about 62% of the $PM_{2.5}$ mass. The

**429** Shandong emissions generally contribute more $PM_{2.5}$ mass than the other five provinces

**430** when the air quality is polluted, with the $PM_{2.5}$ contribution exceeding 10%. The background

**431** source accounts for more than 20% with the air quality being excellent or good. In Shandong,

**432** the local emissions dominate the air quality, generally contributing more than 60% of the

**433** $PM_{2.5}$ mass. The total $PM_{2.5}$ contribution of emissions from Beijing, Tianjin, Hebei, Henan,

**434** and Shanxi is less than 30%, and $PM_{2.5}$ contributions of background source range from 10%

**435** to 15% under different pollution levels. The air quality in Shanxi is mainly decided by local

**436** emissions, with the $PM_{2.5}$ contribution of 58.7%, 57.8%, 43.8%, and 47.7% when the air

**437** quality is excellent, good, slightly, and moderately polluted, respectively. Hebei and Henan

**438** emissions contribute more than 10% and 15% of the $PM_{2.5}$ mass in Shanxi, when the air

**439** quality is slightly and moderately polluted. The $PM_{2.5}$ contribution of background source is

notable, generally exceeding 20%.

441       Table 4, Table 5 and Figure 10 further show the average contribution of local and

non-local emissions to the aerosol species in Beijing, Tianjin, Hebei, Henan, Shandong, and
Shanxi during the episode. Interestingly, the local emissions dominate the elemental carbon
(EC) and POA in Beijing, with a contribution of 61.1% and 64.1%. Hu et al. (2015) have also
revealed that local emissions constitute the major source of POA in Beijing, particularly
during wintertime. Additionally, local emissions also account for around 32% of the SOA in
Beijing, and the high organic aerosol contribution is likely caused by emissions of large
amounts of vehicles in Beijing. Except for EC and POA, non-local emissions dominate the
aerosol species concentration in Beijing, with contributions exceeding 60%, especially for
sulfate and nitrate in which the contribution of non-local emissions is more than 90% (Figure
10). Ying et al. (2014) have shown that the inter-regional transport of air pollutants plays an
important role in the secondary aerosols formation during the polluted episode in China. Sun
et al. (2016) have also demonstrated that the secondary aerosol formed on a regional scale
dominates the aerosol compositions during the haze episode, with an average of 67%.
Apparently, the impact of Hebei emissions on PM pollution in Beijing is the most significant,
with the nitrate and ammonium contribution exceeding 40% (Table 4). Except for EC and
POA, contributions of background source to the aerosol species in Beijing is generally more
than 10%. It is worth noting that the nitrate contribution of the background source is 32.1%,
which is caused by the slow oxidation of $NO_2$ during wintertime.

460       In Tianjin, the non-local emissions play a dominant role in concentrations SOA, sulfate,

nitrate, and ammonium, with contributions of 73.6%, 68.6%, 88.7%, and 71.3%, and also
account for 48.1% and 50.7% of the EC and POA mass, respectively. In general, Hebei
emissions constitute the most important contributor of aerosol species in the non-local
sources, followed by Shandong emissions. In Hebei, the local emissions determine the levels
of EC, POA, sulfate, and ammonium, with contributions of 73.8%, 63.0%, 64.3%, and 67.4%,
respectively. The SOA mass is mainly contributed by local (49.4%) and Shandong (16.7%)
emissions, and background sources (11.6%). However, the non-local emissions dominate the
nitrate mass in Hebei, with the contribution of 78.7%, most of which is from Henan (11.4%),
Shandong (14.6%), Shanxi (10.8%), and background sources (22.9%). Except for sulfate, the
aerosol species in Henan are generally controlled by local emissions, with contributions
varying from 45% to 65%. The sulfate contribution of non-local emissions is 83.2%, mainly
contributed by Hebei (16.7%), Shandong (14.9%), Shanxi (12.1%), and background (22%).
The local emissions contribute about 60~80% of the aerosol species mass in Shandong,
except nitrate aerosols, which are dominated by non-local emissions with a contribution of
75.1%. More than 60% of EC, POA, sulfate and ammonium in Shanxi are formed from local
emissions, but the non-local emissions are the dominant contributor to SOA and nitrate
concentrations.

**4    Summary and conclusions**
We have developed a source-oriented WRF-Chem model, treating the precursors of
aerosols from different sources and their corresponding reaction products as different species
and tracked independently in chemical, physical, and dynamic processes. The model is used
to evaluate contributions of local and non-local emissions to the PM pollution in the NCP,
including Beijing, Tianjin, Hebei, Henan, and Shandong, as well as the adjacent province on
the west (Shanxi) during a persistent and severe haze episode from 05 December 2015 to 04
January 2016. The model exhibits good performance in predicting the temporal variation and
spatial distribution of air pollutants in the NCP and also reasonably simulates the aerosol
species against measurements in Beijing.
In this study, the source-oriented WRF-Chem model is also used to mark the precursors
emitted from residential, transportation, industry, power, and agriculture sectors, respectively,
to evaluate the contribution of anthropogenic emissions to the $PM_{2.5}$ concentration in the NCP.
The average contribution of residential emissions to the $PM_{2.5}$ level is the most significant,
with a maximum exceeding 100 µg m$^{-3}$ during the study episode (Figure S21). In addition,
the contribution of industrial emissions to $PM_{2.5}$ concentration in the NCP also varies from 10
from 100 µg m$^{-3}$ during the study episode. Therefore, more attention should be paid to
residential and industrial sectors to control the air pollution in a more cost-effective way. As
two megacities in the NCP, Beijing and Tianjin have made great efforts to decrease local
emissions of air pollutants since 2013, such as replacing residential coal use with gas and
electricity, elevating vehicle emissions standards, and phasing out high-emitting industries
(Zhang et al., 2019). However, heavy PM pollution events still occur in the two cities, which
is mainly a result of trans-boundary transport of air pollutants. Simulations of the
source-oriented WRF-Chem model reveal that, on average, local and non-local emissions
contribute 36.3% and 63.7% of the $PM_{2.5}$ mass in Beijing during the episode. When the air
quality is excellent or good in terms of hourly $PM_{2.5}$ concentrations, the local emissions
contribute more than 50% to the $PM_{2.5}$ mass, dominating Beijing's air quality. However, with
deterioration of Beijing's air quality from being slightly to severely polluted, the $PM_{2.5}$
contribution of local emissions decreases from 48.7% to 25.1%, indicating the significant
contribution of trans-boundary transport to the PM pollution in Beijing. The non-local
emissions account for 67.3% of the $PM_{2.5}$ mass in Tianjin and the contribution exceeds 70%
when the air quality is severely polluted. The $PM_{2.5}$ concentrations in three industrialized
provinces, Hebei, Shandong, and Henan in the NCP, are generally dominated by the local
emissions under different pollution levels, particularly in Shandong with the $PM_{2.5}$
contribution of local emissions exceeding 60%. The contribution of residential and industrial
emissions to the $PM_{2.5}$ concentration in Hebei, Shandong, and Henan is the most obvious
(Figure S21). Therefore, efficient emission mitigations of air pollutants in the three provinces
need to be carried out continuously to lower PM levels. However, when severe PM pollution
occurs, the $PM_{2.5}$ contribution of local emissions in Hebei and Henan decreases considerably.
The impact of Shanxi's emissions on $PM_{2.5}$ concentrations in the NCP is generally not
significant.

520          The primary aerosol species, such as EC and POA, are generally controlled by local

emissions with the average contribution ranging from about 50% to 85% in the six provinces.
However, the SA of secondary aerosols shows large differences during the episode, with
more evident regional characteristics. Local emissions contribute more than 60% of the SOA
mass in Shandong, 40~50% in Hebei, Henan and Shanxi, and around 30% in Beijing and
Tianjin. The sulfate contribution of local emissions is significant in Hebei, Shandong and
Shanxi, exceeding 60%, but less than 10% in Beijing. Except in Henan, local emissions do
not play an important role in the nitrate formation, with contributions less than 30%, and
most of the nitrate aerosols are produced during trans-boundary transport of its precursors.
Ammonium aerosols in Beijing and Tianjin are mainly determined by non-local emissions,
with the contribution of around 70%. Local emissions in the other four provinces account for
around 60% of the ammonium mass.

532          The developed source-oriented model is mainly used in this study to quantitatively

evaluate the local and non-local contributions to the PM pollution in the NCP. A recent study
(Huang et al., 2020) has demonstrated that, absorption aerosols contributed by cross-regional
transport from the Yangtze River Delta (YRD) to the upper PBL in the NCP induce the
aerosol-PBL interaction and further lead to the suppressed PBL height, notable reduction of
temperature and a substantial enhancement of relative humidity, favoring secondary aerosol
production and aggravation of air pollution in the NCP. In this study, a sensitivity study
without BC transported from the south of 32°N is conducted to analyze the contribution of
the effect of cross-regional transport of air pollutants on local meteorological conditions
during the selected simulated episode. The temperature and PBL height decreases in the NCP
caused by the BC transported from the south are not significant, with a maximum of 0.04 °C
and 1.6%, and the increase of relative humidity just varies from -0.2% to 0.1% (Figure S22).
Therefore, the aerosol-PBL interaction induced by the trans-boundary transport of absorption
aerosols can not be observed in this study. In the future, more typical air pollution episodes
need to be simulated to quantify the impact of regional transport of absorption aerosols on
meteorological conditions.
In order to reduce PM pollution, the cooperation to carry out strict emission mitigation
measures is critical for all provinces, especially with regard to Beijing and Tianjin. In Beijing
and Tianjin, reducing direct emissions of primary aerosols, such as EC and POA, constitutes
the priority, and more efforts need to be made to reduce local emissions of air pollutants in
Hebei, Henan, Shandong, and Shanxi.

*Competing interests.* The authors declare no competing financial interest.

*Data availability.* The real-time $PM_{2.5}$, $O_3$, $NO_2$, $SO_2$ and CO observations are accessible for
the public on the following website: http://106.37.208. 233:20035/ (last access: 24 November
2019) (China MEP, 2013a). One can also access the historic profile of observed ambient
pollutants by visiting http://www.aqistudy.cn/ (last access: 24 November 2019) (China MEP,
2013b).

*Author contribution.* Guohui Li, as the contact author, provided the ideas and financial
support, developed the model code, verified the conclusions, and revised the paper. Jiarui Wu
conducted a research, designed the experiments, performed the simulation, processed the data,
prepared the data visualization, and prepared the manuscript with contributions from all
authors. Naifang Bei validated the model performance, analyzed the study data, and reviewed
the manuscript. Yuan Wang validated the model performance, verified the results and
provided the critical reviews. Suixin Liu provided the data and the primary data process, and
reviewed the manuscript. Xia Li, Lang Liu, Ruonan Wang, Jiaoyang Yu, Tianhao Le, and
Min Zuo analyzed the initial simulation data, visualized the model results and reviewed the
paper. Zhenxing Shen, Junji Cao and Xuexi Tie provided critical reviews pre-publication
stage.

*Acknowledgements.* This work is financially supported by the Strategic Priority Research
Program of Chinese Academy of Sciences (XDB40030203), the National Key R&D Plan
(Quantitative Relationship and Regulation Principle between Regional Oxidation Capacity of
Atmospheric and Air Quality (2017YFC0210000)), and National Research Program for Key
Issues in Air Pollution Control (DQGG0105).

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

**842**

**843**     Table 1 WRF-Chem model configurations.

| Region | North China Plain |
|---|---|
| Simulation period | 05 December 2015 to 04 January 2016 |
| Domain size | 300 × 300 |
| Domain center | 38°N, 116°E |
| Horizontal resolution | 6 km × 6 km |
| Vertical resolution | 35 vertical levels with a stretched vertical grid with spacing ranging from 30 m near the surface, to 500 m at 2.5 km and 1 km above 14 km |
| Microphysics scheme | WSM 6-class graupel scheme (Hong and Lim, 2006) |
| Cumulus scheme | Grell-Devenyi ensemble scheme (Grell and Devenyi, 2002) |
| Boundary layer scheme | MYJ TKE scheme(Janjić, 2002) |
| Surface layer scheme | MYJ surface scheme (Janjić, 2002) |
| Land-surface scheme | Unified Noah land-surface model (Chen and Dudhia, 2001) |
| Longwave radiation scheme | Goddard longwave scheme (Chou and Suarez, 2001) |
| Shortwave radiation scheme | Goddard shortwave scheme (Chou and Suarez, 1999) |
| Meteorological boundary and initial conditions | NCEP 1°×1° reanalysis data |
| Chemical initial and boundary conditions | MOZART 6-hour output (Horowitz et al., 2003) |
| Anthropogenic emission inventory | Developed by Zhang et al. (2009) and Li et al. (2017), 2012 base year, and SAPRC-99 chemical mechanism |
| Biogenic emission inventory | Online MEGAN model developed by Guenther et al. (2006) |
| Model spin-up time | 4 days and 4 hours (Simulations starting time: 12:00 UTC on November 30, 2015) |

**844**
**845**
**846**
**847**
**848**

Table 2 Average PM$_{2.5}$ contributions (%) in Beijing, Tianjin, and Hebei under different
pollution levels from local, the other five provinces, and background source from 05
December 2015 to 04 January 2016.

| Pollution Level (μg m$^{-3}$) | 0-35 | 35-75 | 75-115 | 115-150 | 150-250 | >250 |
|---|---|---|---|---|---|---|
| **Beijing** | | | | | | |
| Beijing | 56.8±12.8 | 55.0 ±13.8 | 48.7±14.5 | 40.5±11.3 | 35.4±11.5 | 25.1±6.3 |
| Tianjin | 1.1±0.7 | 3.7±3.0 | 5.2±3.7 | 9.3±5.3 | 8.0±3.9 | 8.0±1.1 |
| Hebei | 16.9±4.3 | 20.4±7.7 | 24.8±8.7 | 28.4±6.6 | 28.4±7.4 | 21.2±3.2 |
| Henan | 1.1±1.0 | 1.2±1.1 | 1.8±1.2 | 1.4±1.4 | 3.4±2.0 | 6.2±1.7 |
| Shandong | 1.1±1.0 | 1.2±1.2 | 2.0±1.8 | 2.4±2.7 | 7.1±5.8 | 16.4±6.6 |
| Shanxi | 2.2±1.5 | 3.0±2.1 | 3.8±1.9 | 2.9±2.1 | 4.8±1.8 | 2.5±1.9 |
| Background | 20.8±10.0 | 15.4±8.3 | 13.8±7.3 | 15.1±5.8 | 12.8±5.5 | 20.6±3.9 |
| **Tianjin** | | | | | | |
| Beijing | 21.6±12.1 | 7.8±7.7 | 5.7±4.8 | 5.9±3.9 | 7.8±5.3 | 8.8±6.3 |
| Tianjin | 36.5±11.3 | 44.9±12.7 | 41.3±14.1 | 37.0±11.7 | 29.6±9.6 | 27.5±7.2 |
| Hebei | 23.1±5.1 | 28.3±8.3 | 30.4±10.1 | 31.7±10.4 | 30.6±9.8 | 27.8±5.4 |
| Henan | 0.8±0.4 | 1.1±1.3 | 1.3±1.3 | 2.1±1.4 | 3.7±1.7 | 6.7±3.6 |
| Shandong | 0.8±0.5 | 2.0±2.1 | 3.6±3.4 | 6.2±6.9 | 13.9±11.1 | 18.0±11.1 |
| Shanxi | 0.8±0.5 | 1.3±1.3 | 1.6±1.3 | 2.3±1.2 | 3.0±1.3 | 4.1±1.2 |
| Background | 16.5±9.0 | 14.6±9.5 | 16.0±10.7 | 14.9±8.6 | 11.4±7.2 | 7.1±5.6 |
| **Hebei** | | | | | | |
| Beijing | 4.1±1.5 | 5.7±2.1 | 5.7±2.2 | 6.2±2.0 | 5.0±1.8 | 5.8±1.3 |
| Tianjin | 2.7±1.1 | 5.2±2.7 | 5.3±2.2 | 5.5±1.2 | 5.4±1.5 | 6.7±0.7 |
| Hebei | 65.8±11.2 | 60.9±10.3 | 53.8±8.0 | 50.3±7.0 | 45.2±6.0 | 49.0±4.0 |
| Henan | 0.9±0.4 | 3.1±2.2 | 5.4±3.6 | 5.8±3.8 | 9.3±3.8 | 6.7±0.8 |
| Shandong | 0.9±0.5 | 5.4±3.3 | 11.3±5.1 | 12.7±5.1 | 18.0±4.0 | 18.6±2.7 |
| Shanxi | 6.4±3.2 | 4.4±2.3 | 5.4±1.6 | 5.6±1.4 | 5.7±1.1 | 5.1±0.7 |
| Background | 19.2±8.3 | 15.2±5.1 | 13.1±4.9 | 13.9±5.5 | 11.3±5.6 | 8.2±0.5 |

Table 3 Same as Table 2, but for Henan, Shandong, and Shanxi.

| Pollution Level (µg m$^{-3}$) | 0-35 | 35-75 | 75-115 | 115-150 | 150-250 | >250 |
|---|---|---|---|---|---|---|
| **Henan** | | | | | | |
| Beijing | 0.1±0.1 | 1.2±1.2 | 1.5±1.3 | 2.2±1.4 | 2.4±0.9 | 2.7±0.5 |
| Tianjin | 0.2±0.1 | 1.2±1.2 | 1.5±1.3 | 2.3±1.3 | 2.3±1.0 | 3.1±0.7 |
| Hebei | 2.4±1.3 | 4.1±2.1 | 6.9±4.7 | 9.2±5.1 | 12.1±5.8 | 18.3±2.0 |
| Henan | 55.2±15.0 | 55.3±11.1 | 55.3±12.3 | 50.1±10.2 | 45.5±9.4 | 38.0±6.3 |
| Shandong | 2.8±1.3 | 6.5±7.3 | 11.3±8.6 | 13.5±6.6 | 13.1±5.5 | 20.0±4.0 |
| Shanxi | 12.9±5.5 | 8.2±3.3 | 4.7±2.9 | 5.0±3.0 | 5.0±2.4 | 5.9±0.9 |
| Background | 26.3±8.8 | 23.5±6.4 | 18.8±7.1 | 17.7±7.9 | 19.7±8.0 | 11.9±3.2 |
| **Shandong** | | | | | | |
| Beijing | 4.2±1.3 | 1.8±1.4 | 2.7±1.4 | 2.4±1.5 | 3.0±1.6 | 2.2±0.5 |
| Tianjin | 3.8±1.1 | 2.0±1.3 | 3.2±1.8 | 2.4±1.7 | 3.3±1.9 | 2.2±0.5 |
| Hebei | 11.8±8.9 | 11.5±6.8 | 9.6±5.4 | 5.5±3.5 | 9.6±6.1 | 5.2±2.4 |
| Henan | 3.5±1.4 | 3.5±1.2 | 4.4±1.8 | 6.1±3.5 | 8.6±4.1 | 10.1±4.6 |
| Shandong | 59.2±16.0 | 64.2±13.3 | 62.3±16.4 | 69.7±12.9 | 61.7±11.9 | 66.5±11.7 |
| Shanxi | 3.8±0.8 | 2.6±1.8 | 2.8±1.7 | 2.5±1.7 | 3.6±1.5 | 3.4±0.9 |
| Background | 13.8 ±8.7 | 14.4±7.1 | 15.2±8.6 | 11.3±9.5 | 10.3±9.6 | 10.3±6.3 |
| **Shanxi** | | | | | | |
| Beijing | 1.3±1.5 | 1.6±0.9 | 1.6±1.3 | 1.2±0.3 | / | / |
| Tianjin | 1.3±1.5 | 1.2±0.7 | 1.4±1.3 | 1.0±0.2 | / | / |
| Hebei | 1.8±1.6 | 7.2±5.2 | 10.3±6.3 | 10.0±2.1 | / | / |
| Henan | 1.8±1.6 | 7.9±4.6 | 18.0±8.1 | 17.7±3.8 | / | / |
| Shandong | 1.3±1.5 | 1.9±1.3 | 3.4±2.0 | 2.7±0.4 | / | / |
| Shanxi | 58.7±13.3 | 57.8±11.1 | 43.8±9.1 | 47.7±1.7 | / | / |
| Background | 33.6±13.6 | 22.3±8.9 | 21.5±7.1 | 19.7±2.7 | / | / |


Table 4 Average aerosol constituent contributions (%) in Beijing, Tianjin, and Hebei from
local, the other five, and background source from 05 December 2015 to 04 January 2016.

| Species | EC | POA | SOA | Sulfate | Nitrate | Ammonium |
|---|---|---|---|---|---|---|
| **Beijing** | | | | | | |
| Beijing | 61.1±14.3 | 64.1±14.3 | 31.9±15.8 | 9.8±6.7 | 10.0±4.5 | 32.5±12.7 |
| Tianjin | 5.1±5.0 | 7.0±5.5 | 8.5±6.6 | 7.8±7.0 | 8.6±4.6 | 7.5±5.2 |
| Hebei | 24.9±9.6 | 19.0±7.6 | 29.1±11.4 | 48.0±22.0 | 19.1±6.6 | 40.8±10.0 |
| Henan | 0.6±1.0 | 0.7±0.9 | 2.1±2.7 | 3.9±3.3 | 8.6±4.4 | 2.5±2.2 |
| Shandong | 2.3±3.0 | 3.2±3.9 | 7.1±7.2 | 9.8±6.6 | 10.5±5.5 | 5.0±4.6 |
| Shanxi | 1.3±1.6 | 2.1±1.5 | 3.5±2.3 | 7.8±6.6 | 11.0±4.8 | 1.7±1.1 |
| Background | 4.6±6.0 | 3.9±4.3 | 17.7±23.1 | 12.7±35.9 | 32.1±24.7 | 10.0±8.2 |
| **Tianjin** | | | | | | |
| Beijing | 5.3±5.6 | 7.1±7.2 | 13.8±9.0 | 1.1±1.4 | 10.2±5.4 | 3.4±3.7 |
| Tianjin | 51.9±11.2 | 49.3±14.7 | 26.4±14.6 | 31.4±12.9 | 11.3±5.5 | 28.7±11.6 |
| Hebei | 23.7±9.3 | 18.7±8.6 | 23.8±11.2 | 27.7±23.0 | 19.4±6.7 | 31.5±11.3 |
| Henan | 2.3±1.8 | 2.8±1.7 | 5.2±3.1 | 6.5±9.1 | 11.1±5.4 | 6.8±3.5 |
| Shandong | 9.8±6.3 | 15.3±10.8 | 20.7±13.5 | 20.3±21.2 | 16.7±7.3 | 17.5±10.4 |
| Shanxi | 1.3±1.0 | 1.5±1.0 | 2.6±1.4 | 4.4±4.8 | 10.6±5.4 | 0.8±0.5 |
| Background | 5.9±6.0 | 5.3±6.5 | 7.5±7.0 | 8.6±9.1 | 20.6±28.6 | 11.2±9.7 |
| **Hebei** | | | | | | |
| Beijing | 4.4±2.6 | 7.2±3.7 | 6.0±3.2 | 0.8±0.5 | 9.4±3.7 | 2.4±1.1 |
| Tianjin | 3.7±2.0 | 4.8±2.3 | 5.3±2.7 | 3.1±2.6 | 9.5±4.2 | 3.2±1.4 |
| Hebei | 73.8±9.3 | 63.0±11.5 | 49.4±10.0 | 64.3±32.9 | 21.3±6.0 | 67.4±10.3 |
| Henan | 4.1±3.2 | 5.9±3.8 | 7.8±4.1 | 9.2±14.2 | 11.4±4.3 | 9.3±5.5 |
| Shandong | 6.5±6.0 | 11.4±8.6 | 16.7±9.3 | 12.6±16.1 | 14.6±5.6 | 9.7±6.3 |
| Shanxi | 2.4±2.1 | 3.0±2.7 | 3.2±2.6 | 5.0±2.4 | 10.8±4.1 | 1.2±0.7 |
| Background | 5.0±4.2 | 4.8±3.5 | 11.6±6.6 | 4.9±2.2 | 22.9±18.6 | 6.9±4.9 |

**873** Table 5 Same as Table 4, but for Henan, Shandong, and Shanxi.

**874**

| Species | EC | POA | SOA | Sulfate | Nitrate | Ammonium |
|---|---|---|---|---|---|---|
| **Henan** | | | | | | |
| Beijing | 0.6±0.9 | 0.5±1.0 | 1.1±1.0 | 8.7±2.6 | 0.2±0.2 | 0.6±0.4 |
| Tianjin | 0.7±0.9 | 0.6±0.7 | 0.8±0.7 | 8.7±2.8 | 0.4±0.2 | 0.7±0.4 |
| Hebei | 16.5±9.0 | 11.9±6.6 | 13.9±5.7 | 16.7±19.0 | 14.4±6.1 | 16.5±7.2 |
| Henan | 56.5±12.6 | 59.2±13.3 | 45.0±13.6 | 16.8±14.3 | 64.3±10.3 | 56.5±11.8 |
| Shandong | 8.6±6.1 | 12.1±9.6 | 14.4±9.8 | 14.9±13.2 | 7.9±6.6 | 8.6±5.9 |
| Shanxi | 5.4±4.7 | 6.1±4.9 | 4.9±5.2 | 12.1±11.2 | 2.0±3.2 | 5.4±1.6 |
| Background | 11.7±8.3 | 9.5±6.6 | 19.8±11.7 | 22.0±16.3 | 10.8±6.0 | 11.7±6.4 |
| **Shandong** | | | | | | |
| Beijing | 1.0±1.4 | 1.0±1.5 | 2.1±2.4 | 0.2±0.4 | 10.1±5.6 | 0.5±0.6 |
| Tianjin | 1.1±1.3 | 1.0±1.4 | 1.4±1.7 | 1.0±2.4 | 10.5±5.7 | 0.8±0.9 |
| Hebei | 7.5±8.8 | 4.5±5.8 | 6.5±7.5 | 7.1±15.5 | 16.5±5.2 | 7.3±7.9 |
| Henan | 5.1±3.7 | 5.1±3.2 | 7.9±4.3 | 8.7±17.8 | 13.8±5.3 | 10.2±4.2 |
| Shandong | 71.9±14.5 | 78.2±11.8 | 60.4±17.0 | 68.3±18.8 | 24.9±9.6 | 62.5±14.0 |
| Shanxi | 1.5±1.4 | 1.3±1.1 | 2.0±1.7 | 3.4±3.44 | 11.7±5.7 | 0.7±0.5 |
| Background | 11.8±9.1 | 8.9±6.5 | 19.6±12.5 | 11. 3±9.2 | 12.6±18.0 | 18.0±6.0 |
| **Shanxi** | | | | | | |
| Beijing | 0.4±1.0 | 0.4±1.1 | 1.5±2.3 | 0.1±0.3 | 7.1±5.3 | 0.3±0.5 |
| Tianjin | 0.2±0.6 | 0.2±0.6 | 4.0±5.5 | 0.2±1.2 | 6.6±5.3 | 0.3±0.6 |
| Hebei | 5.3±5.8 | 3.2±3.7 | 8.6±7.2 | 5.5±7.1 | 13.7±6.3 | 9.3±8.3 |
| Henan | 4.9±4.3 | 4.4±4.9 | 14.1±10.7 | 10.4±14.3 | 15.3±6.9 | 16.3±13.7 |
| Shandong | 0.7±1.2 | 0.8±1.3 | 2.5±3.1 | 1.3±1.9 | 8.5±5.5 | 1.8±1.8 |
| Shanxi | 79.8±11.4 | 84.1±10.7 | 42.1±9.7 | 74.7±23.9 | 19.4±8.4 | 62.2±15.3 |
| Background | 8.8±5.8 | 6.8±3.2 | 27.1±7.2 | 7.8±3.4 | 29.5±22.6 | 9.7±6.2 |

**875**
**876**
**877**
**878**
**879**

 **Figure Captions**


Figure 1 WRF-Chem simulation domain with topography. The circles represent centers of
cities with ambient monitoring sites, and the size of circles denotes the number of
ambient monitoring sites of cities. The red circle denotes observation site for aerosol
species at the National Center for Nanoscience and Technology (NCNST), Chinese
Academy of Sciences, Beijing.
Figure 2 Conceptual scheme of source apportionment for sulfate aerosols formed from (a)
homogeneous and (b) heterogeneous reactions. *FR*: formation rate; Superscript *i*:
source number; Superscript *T*: total; Subscript *g*: gas phase; Subscript *a*: aerosol phase;
Subscript *aq*: aerosols in cloud water.
Figure 3 Conceptual scheme of source apportionment for nitrate and ammonium aerosols.
Superscript *i*: source number; Superscript *T*: total; Subscript *g*: gas phase; Subscript *a*:
aerosol phase.
Figure 4 Conceptual scheme of source apportionment for organic aerosols formed from (a)
homogeneous and (b) heterogeneous reactions. Superscript *i*: source number;
Superscript *T*: total; Subscripts *j* and *k*: volatility bin number; Subscript *g*: gas phase;
Subscript *a*: aerosol phase. AVOC/BVOC: VOCs emitted from
anthropogenic/biogenic source; ASVOC/BSVOC: SVOC from oxidation of
AVOC/BVOC; OPOG: oxidized POG. PSOA: SOA from oxidation and partitioning
of POA treated as semi-volatile; ASOA/BSOA: SOA from oxidation of
anthropogenic/biogenic VOCs; HSOA: SOA from irreversible uptake of glyoxal and
methylglyoxal on aerosol/cloud surfaces.
Figure 5 Comparison of observed (black dots) and simulated (solid red lines) diurnal profiles
of near-surface hourly mass concentrations of (a) $PM_{2.5}$, (b) $O_3$, (c) $NO_2$, (d) $SO_2$, and
(d) CO averaged at monitoring sites in the NCP from 05 December 2015 to 04
January 2016.
Figure 6 Pattern comparisons of simulated (color contours) vs. observed (colored circles)
near-surface mass concentrations of (a) $PM_{2.5}$, (b) $O_3$, (c) $NO_2$, and (d) $SO_2$ averaged
from 05 December 2015 to 04 January 2016. The black arrows indicate simulated
near-surface winds.
Figure 7 Comparison of measured (black dots) and simulated (black line) diurnal profiles of
submicron aerosol species of (a) POA, (b) SOA, (c) sulfate, (d) nitrate, and (e)
ammonium at NCNST site in Beijing from 05 December 2015 to 04 January 2016.
Figure 8 Spatial distribution of average $PM_{2.5}$ contributions from (a) Beijing, (b) Tianjin, (c)
Hebei, (d) Shandong, (e) Henan, and (f) Shanxi provinces from 05 December 2015 to
04 January 2016.
Figure 9 Average $PM_{2.5}$ contributions (%) in (a) Beijing, (b) Tianjin, (c) Hebei, (d) Henan, (e)
Shandong, and (f) Shanxi from local (red) and non-local (blue) emissions from 05
December 2015 to 04 January 2016 under different pollution levels with error bars.
Figure 10 Average aerosol constituent contributions (%) in (a) Beijing, (b) Tianjin, (c) Hebei,
(d) Henan, (e) Shandong, and (f) Shanxi from local (red) and non-local (blue)
emissions from 05 December 2015 to 04 January 2016 with error bars.

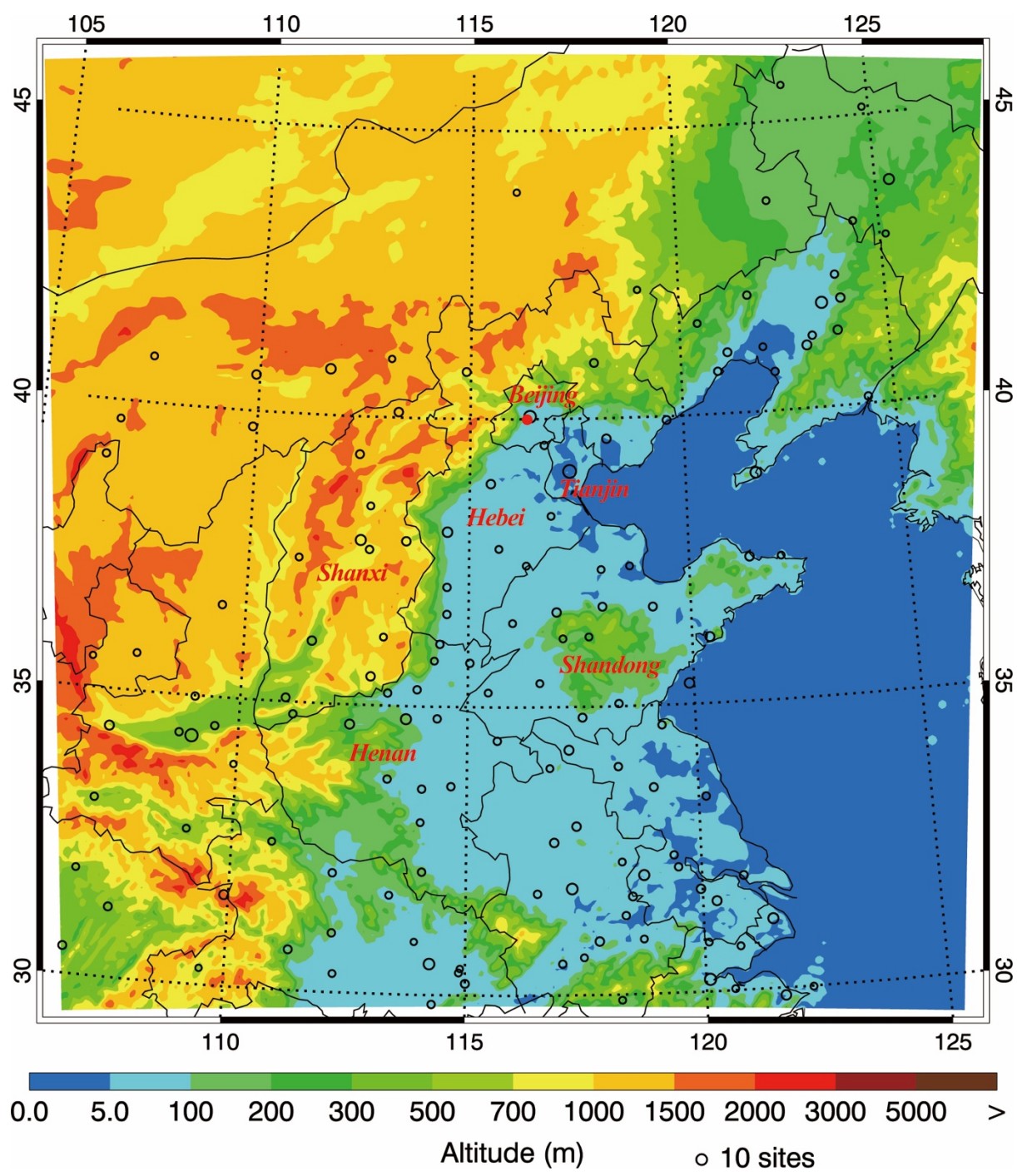



Figure 1 WRF-Chem simulation domain with topography. The circles represent centers of
cities with ambient monitoring sites, and the size of circles denotes the number of ambient
monitoring sites of cities. The red circle denotes observation site for aerosol species at the
National Center for Nanoscience and Technology (NCNST), Chinese Academy of Sciences,
Beijing.





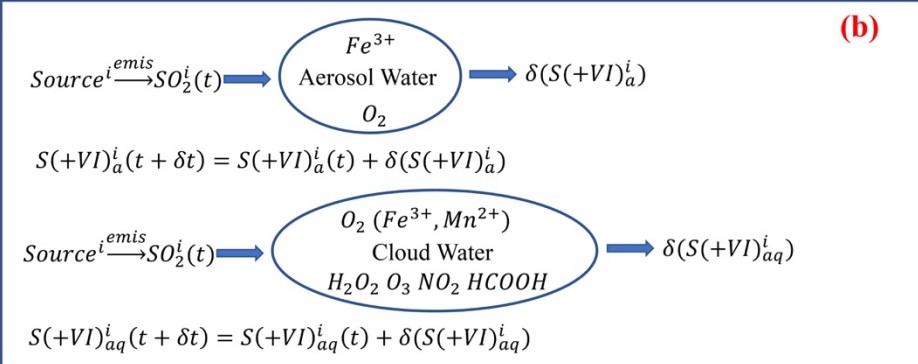

$$Source^i \xrightarrow{emis} SO_2^i + OH/sCI \rightarrow H_2SO_4^i \rightarrow FR(S(+VI)_g^i)\,(t) \qquad \textbf{(a)}$$

$$\sum_i FR(S(+VI)_g^i) = FR(S(+VI)_g^T)(t)$$

$$\sum_i S(+VI)_a^i = S(+VI)_a^T(t)$$

Nucleation Condensation $\rightarrow S(+VI)_a^T(t+\delta t)$

$$S(+VI)_a^i(t+\delta t) = S(+VI)_a^i(t) + \frac{FR(S(+VI)_g^i)(t)}{FR(S(+VI)_g^T)(t)} \times (S(+VI)_a^T(t+\delta t) - S(+VI)_a^T(t))$$

$$\textbf{(b)}$$

$$Source^i \xrightarrow{emis} SO_2^i(t) \rightarrow \begin{array}{c} Fe^{3+} \\ \text{Aerosol Water} \\ O_2 \end{array} \rightarrow \delta(S(+VI)_a^i)$$

$$S(+VI)_a^i(t+\delta t) = S(+VI)_a^i(t) + \delta(S(+VI)_a^i)$$

$$Source^i \xrightarrow{emis} SO_2^i(t) \rightarrow \begin{array}{c} O_2\ (Fe^{3+}, Mn^{2+}) \\ \text{Cloud Water} \\ H_2O_2\ O_3\ NO_2\ HCOOH \end{array} \rightarrow \delta(S(+VI)_{aq}^i)$$

$$S(+VI)_{aq}^i(t+\delta t) = S(+VI)_{aq}^i(t) + \delta(S(+VI)_{aq}^i)$$

Figure 2 Conceptual scheme of source apportionment for sulfate aerosols formed from (a) homogeneous and (b) heterogeneous reactions. *FR*: formation rate; Superscript *i*: source number; Superscript *T*: total; Subscript *g*: gas phase; Subscript *a*: aerosol phase; Subscript *aq*: aerosols in cloud water.

$$Source^i \xrightarrow{emis} NO_X^i + OH/O_3 \rightarrow HNO_3^i \rightarrow N(+V)_g^i \leftrightarrow N(+V)_a^i$$

$$Source^i \xrightarrow{emis} NH_3^i \rightarrow N(-III)_g^i \leftrightarrow N(-III)_a^i$$

$$\sum_i S(+VI)_a^i = S(+VI)_a^T(t)$$

$$\sum_i N(+V)_a^i + N(+V)_g^i = N(+V)^T(t)$$

$$\sum_i N(-III)_a^i + N(-III)_g^i = N(-III)^T(t)$$

ISORROPIA

$$N(+V)_a^T(t+\delta t), N(+V)_g^T(t+\delta t)$$

$$N(-III)_a^T(t+\delta t), N(-III)_g^T(t+\delta t)$$

**Temperature, relative humidity**

$$N(+V)_a^i(t+\delta t) = \frac{N(+V)_a^T(t+\delta t)}{N(+V)_a^T(t+\delta t) + N(+V)_g^T(t+\delta t)} \times (N(+V)_a^i(t) + N(+V)_g^i(t))$$

$$N(-III)_a^i(t+\delta t) = \frac{N(-III)_a^T(t+\delta t)}{N(-III)_a^T(t+\delta t) + N(-III)_g^T(t+\delta t)} \times (N(-III)_a^i(t) + N(-III)_g^i(t))$$

Figure 3 Conceptual scheme of source apportionment for nitrate and ammonium aerosols.
Superscript $i$: source number; Superscript $T$: total; Subscript $g$: gas phase; Subscript $a$:
aerosol phase.

**(a)**

$$Source^i \xrightarrow{emis} POA_j^i \leftrightarrow POG_j^i + OH \rightarrow 1.075 OPOG_{j-1}^i \leftrightarrow PSOA_{j-1}^i \quad j = 1, \dots, 9$$

$$Source^i \xrightarrow{emis} AVOC^i + OH/NO_3/O_3 \rightarrow \sum_{k=1}^{4} \alpha_k ASVOC_k^i \quad ASVOC_k^i \leftrightarrow ASOA_k^i$$

$$Source^i \xrightarrow{emis} BVOC^i + OH/NO_3/O_3 \rightarrow \sum_{k=1}^{4} \beta_k BSVOC_k^i \quad BSVOC_k^i \leftrightarrow BSOA_k^i$$

$$i = 1, \dots, N$$

$POG_j^i(t), POA_j^i(t)$ → **VBS Module** → $POG_j^i(t + \delta t), POA_j^i(t + \delta t)$

$OPOG_j^i(t), PSOA_j^i(t)$ → $OPOG_j^i(t + \delta t), PSOA_j^i(t + \delta t)$

$ASVOC_k^i(t), ASOA_k^i(t)$ → $ASVOC_k^i(t + \delta t), ASOA_k^i(t + \delta t)$

$BSVOC_k^i(t), BSOA_k^i(t)$ → $BSVOC_k^i(t + \delta t), BSOA_k^i(t + \delta t)$

$$j = 1, \dots, 9; k = 1, \dots, 4$$

**(b)**

$$Source^i \xrightarrow{emis} VOC^i + OH/NO_3/O_3 \rightarrow$$

$$Source^i \xrightarrow{emis} \begin{matrix} Glyoxal^i(t) \\ Methylgloxal^i(t) \end{matrix} \xrightarrow{Aerosol/cloud} \delta(HSOA^i)$$

$$HSOA^i(t + \delta t) = HSOA^i(t) + \delta(HSOA^i)$$

Figure 4 Conceptual scheme of source apportionment for organic aerosols formed from (a) homogeneous and (b) heterogeneous reactions. Superscript *i*: source number; Superscript *T*: total; Subscripts *j* and *k*: volatility bin number; Subscript *g*: gas phase; Subscript *a*: aerosol phase. AVOC/BVOC: VOCs emitted from anthropogenic/biogenic source; ASVOC/BSVOC: SVOC from oxidation of AVOC/BVOC; OPOG: oxidized POG. PSOA: SOA from oxidation and partitioning of POA treated as semi-volatile; ASOA/BSOA: SOA from oxidation of anthropogenic/biogenic VOCs; HSOA: SOA from irreversible uptake of glyoxal and methylglyoxal on aerosol/cloud surfaces.

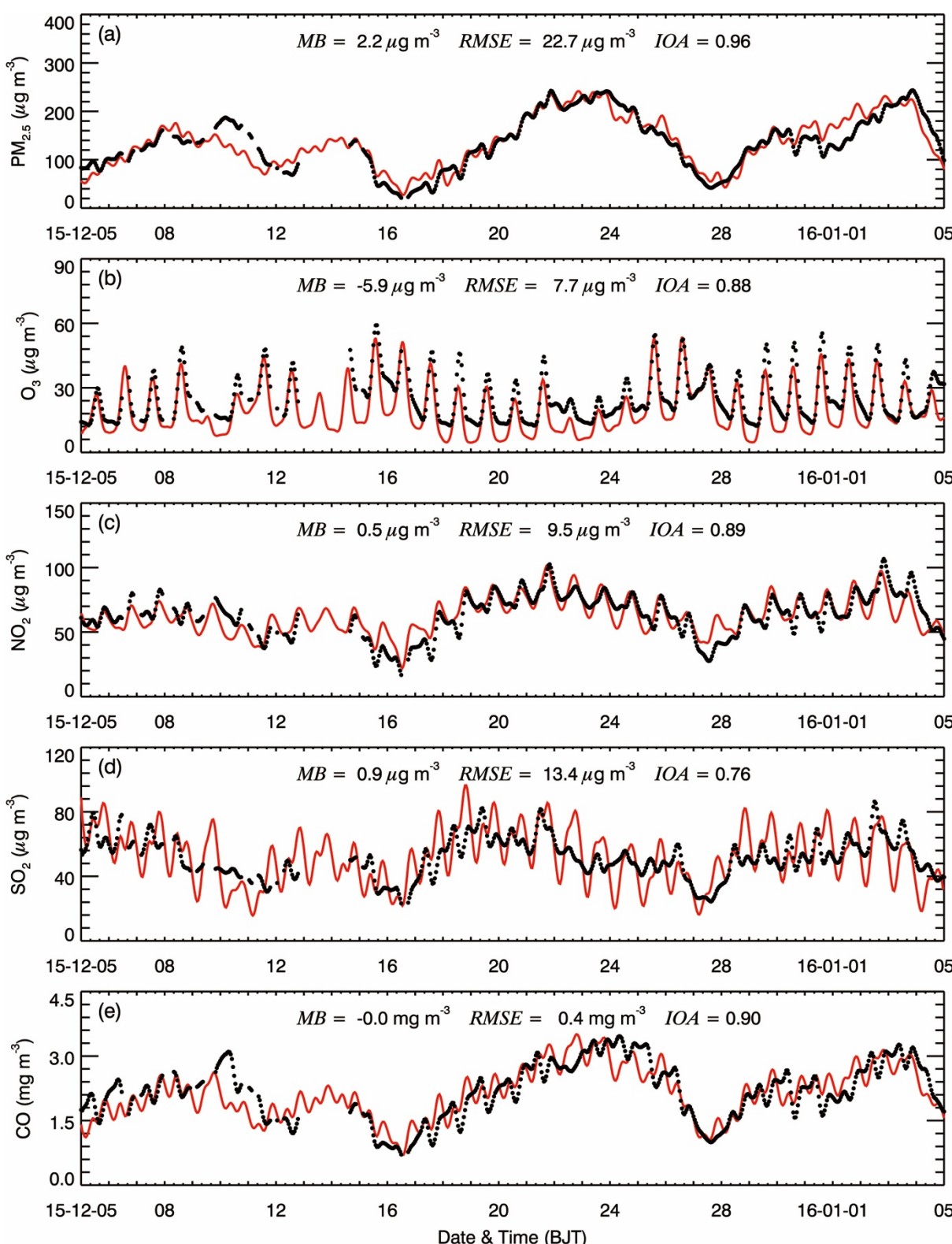



Figure 5 Comparison of observed (black dots) and simulated (solid red lines) diurnal profiles
of near-surface hourly mass concentrations of (a) PM$_{2.5}$, (b) O$_3$, (c) NO$_2$, (d) SO$_2$, and (d) CO
averaged at monitoring sites in the NCP from 05 December 2015 to 04 January 2016.


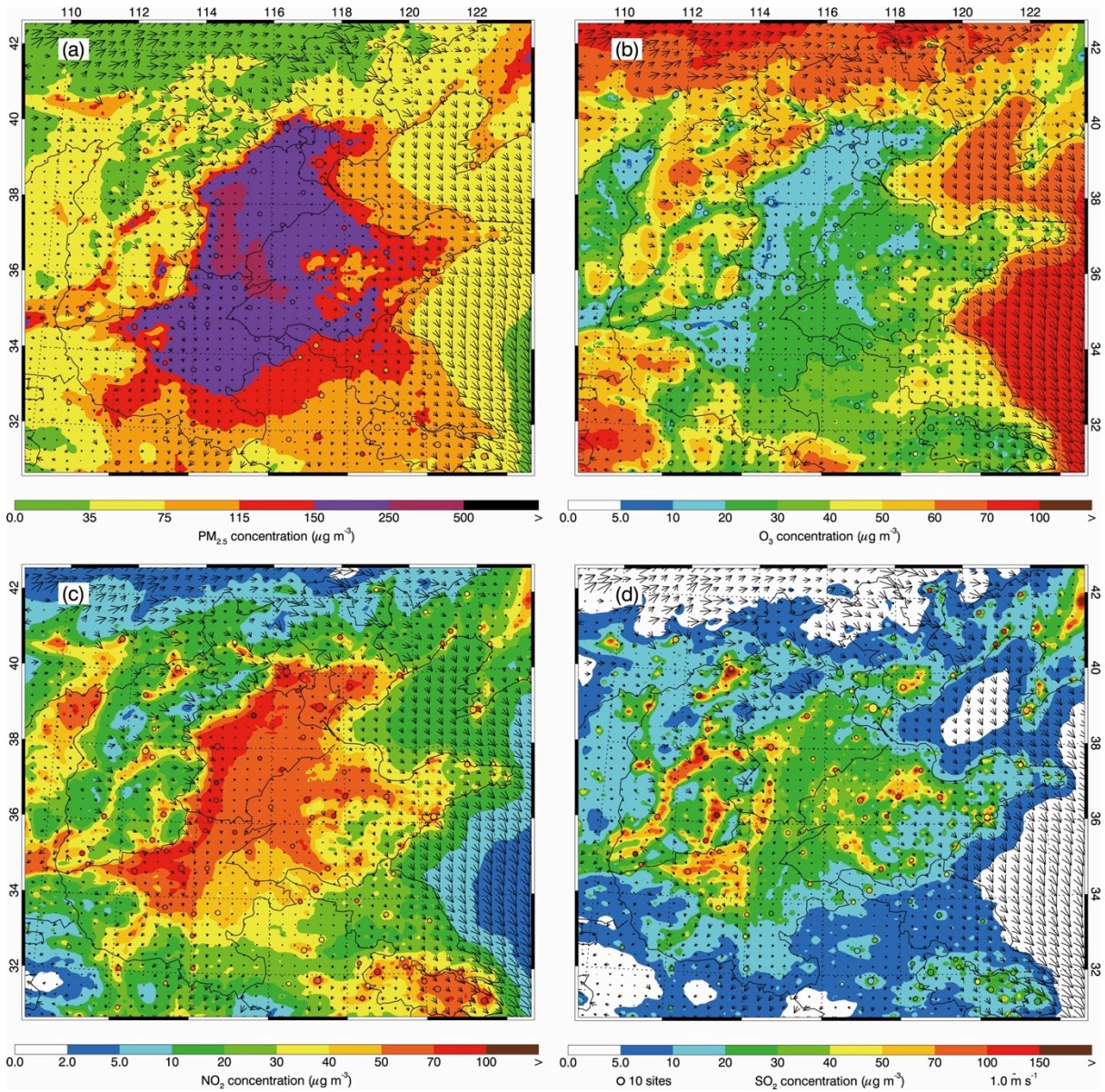

Figure 6 Pattern comparisons of simulated (color contours) vs. observed (colored circles) near-surface mass concentrations of (a) $PM_{2.5}$, (b) $O_3$, (c) $NO_2$, and (d) $SO_2$ averaged from 05 December 2015 to 04 January 2016. The black arrows indicate simulated surface winds.

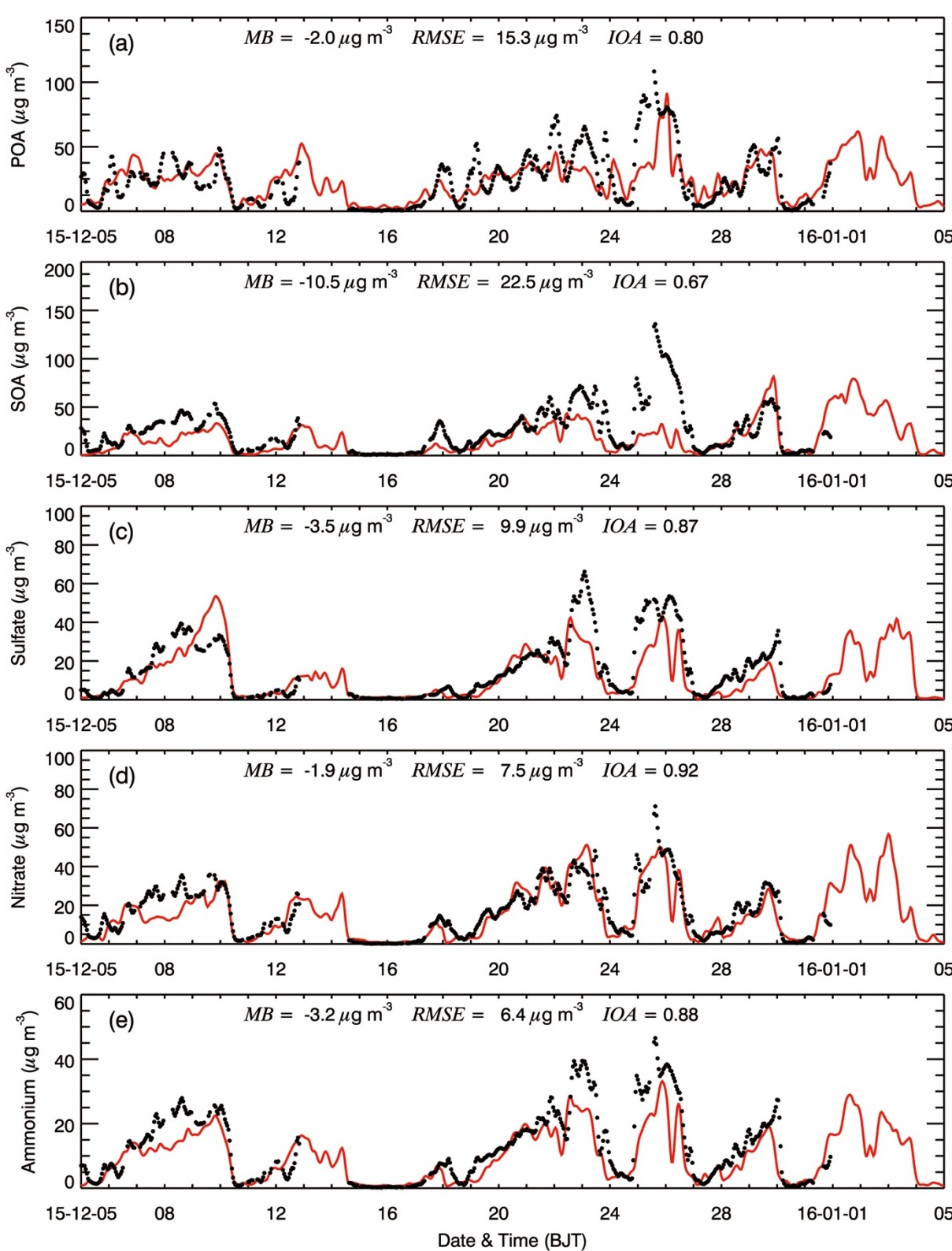

Figure 7 Comparison of measured (black dots) and simulated (black line) diurnal profiles of
submicron aerosol species of (a) POA, (b) SOA, (c) sulfate, (d) nitrate, and (e) ammonium at
NCNST site in Beijing from 05 December 2015 to 04 January 2016.

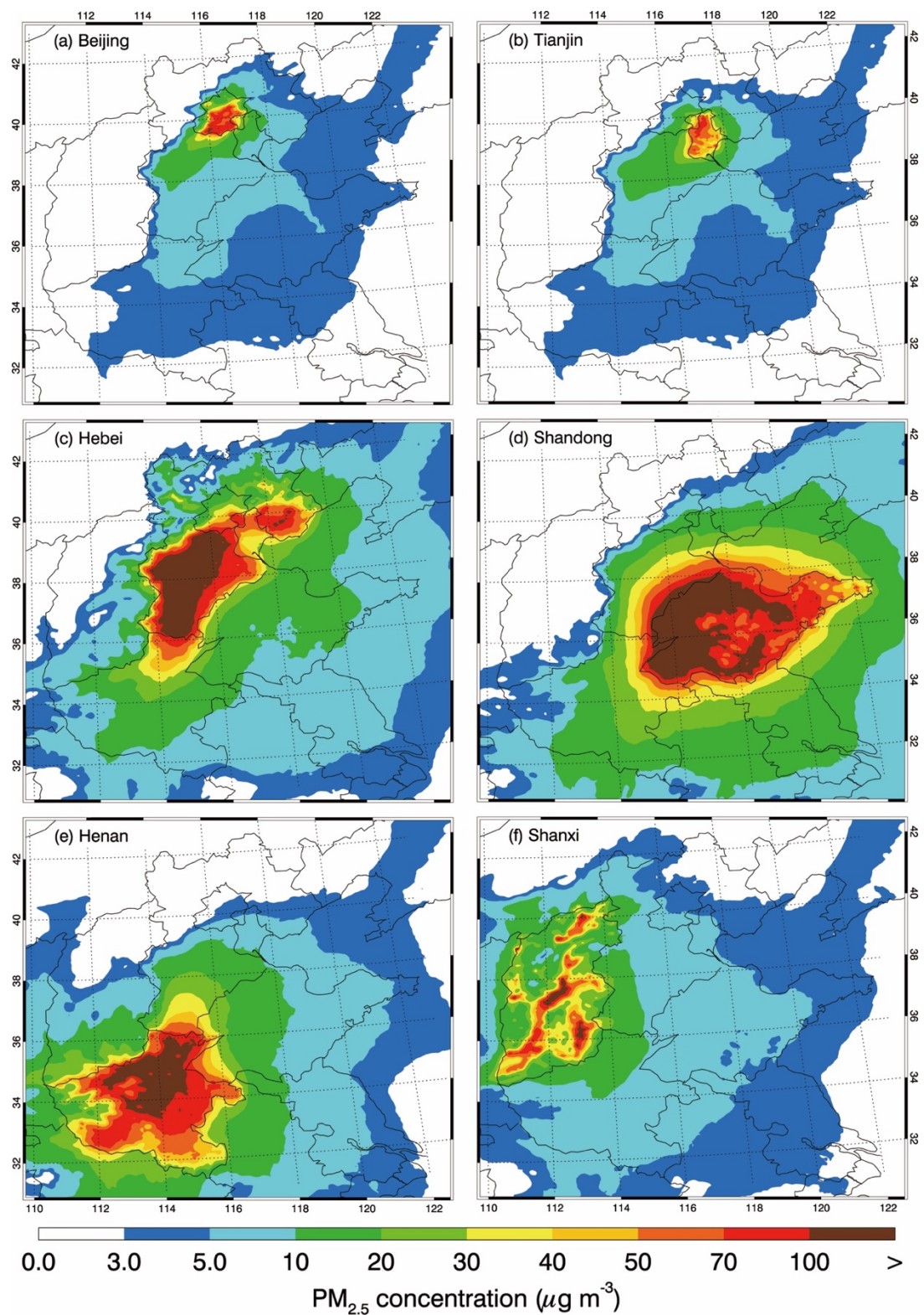

Figure 8 Spatial distribution of average PM$_{2.5}$ contributions from (a) Beijing, (b) Tianjin, (c) Hebei, (d) Shandong, (e) Henan, and (f) Shanxi provinces from 05 December 2015 to 04 January 2016.

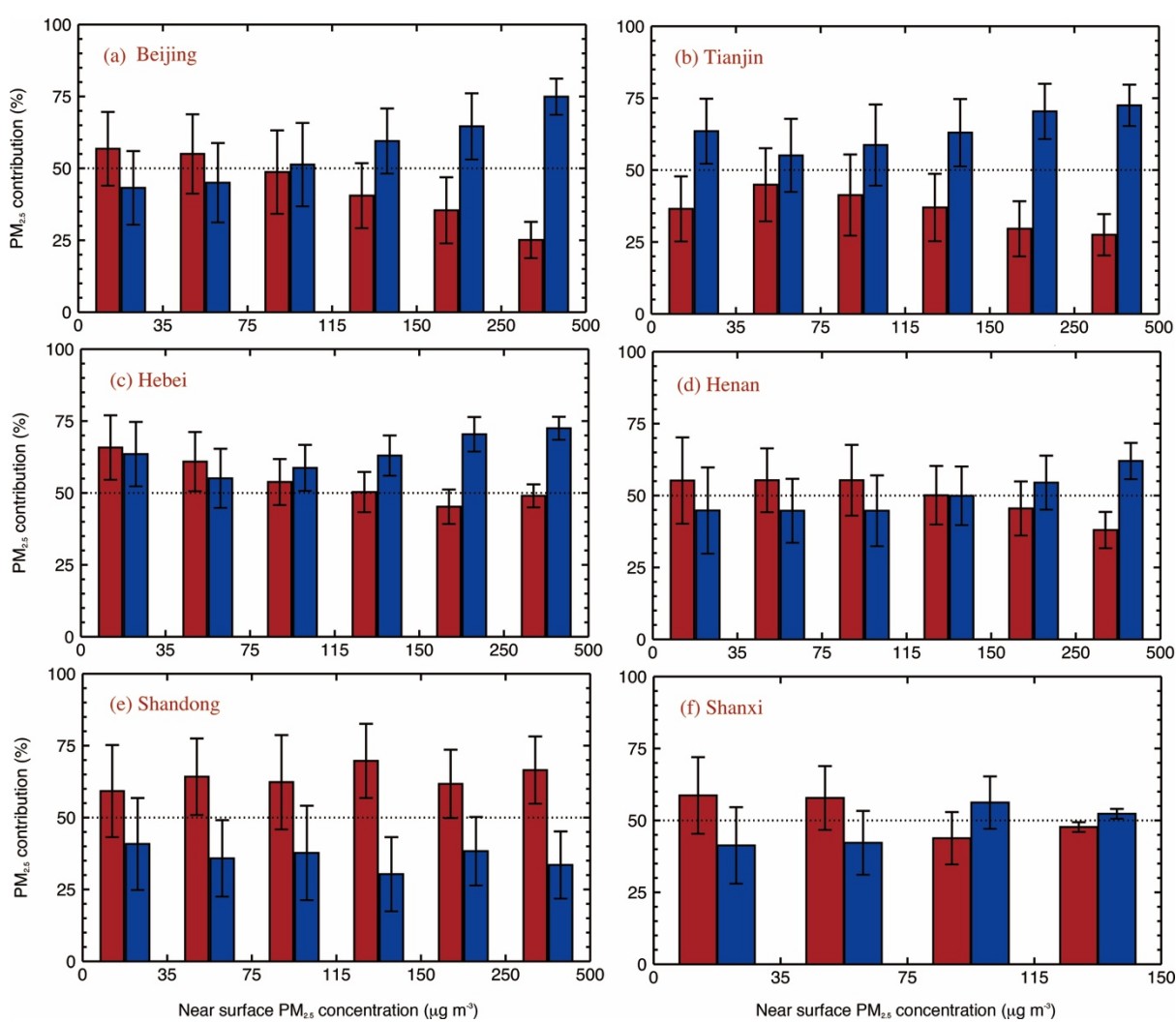

Figure 9 Average PM$_{2.5}$ contributions (%) in (a) Beijing, (b) Tianjin, (c) Hebei, (d) Henan, (e)
Shandong, and (f) Shanxi from local (red) and non-local (blue) emissions from 05 December
2015 to 04 January 2016 under different pollution levels with error bars.

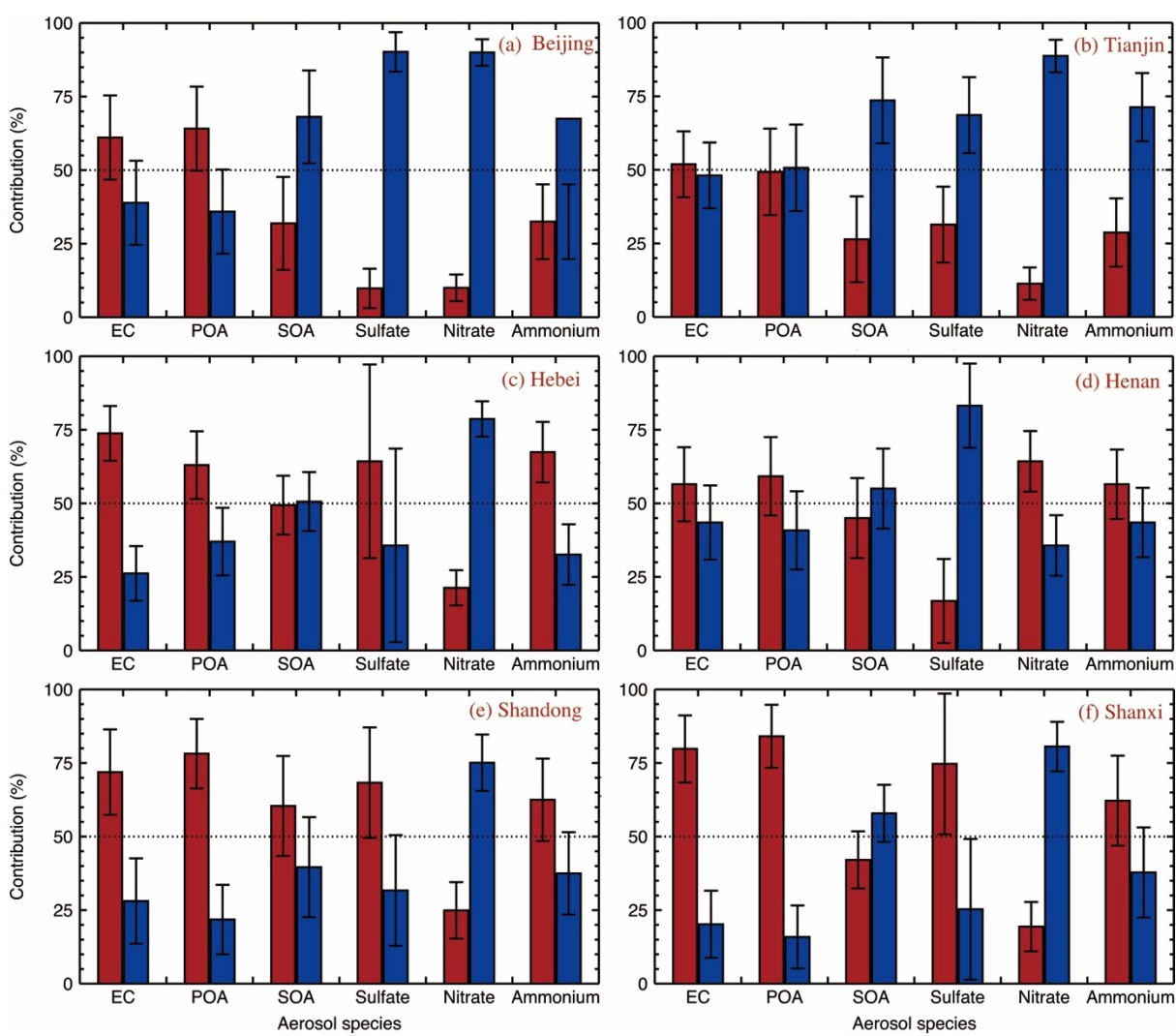

Figure 10 Average aerosol constituent contributions (%) in (a) Beijing, (b) Tianjin, (c) Hebei,
(d) Henan, (e) Shandong, and (f) Shanxi from local (red) and non-local (blue) emissions from
05 December 2015 to 04 January 2016 with error bars.