# Peer review of "Insights into particulate matter pollution in the North China Plain during wintertime: Local contribution or regional transport?"

_Atmospheric Chemistry and Physics, 2020_

## Referee Comment (RC1) · Anonymous Referee #3 · 12 Aug 2020

Source attribution of air pollution is of great importance in emission control policy making. This work developed a source-oriented method in WRF-Chem regional model and applied it in the source appointment of fine particle pollution in the highly-pollution North China Plain region. Based on a one-month simulation using this source-oriented version of the WRF-Chem model, the authors indicated different contributions from local and non-local emissions for Beijing, Tianjin and other provinces and highlight the cooperation among provinces. Overall, this work is well structured but still needs more clarification and some in-depth analysis. Here are some issues that are suggested to be addressed for further improving this work.

[Figure]

Some detailed configurations of model need to be clarified and updated. The definition of source regions needs to be provided in the method. There is no information or figure of the model domain and designation of source regions. See Fig.1 in Hu et al. (2015). Also, technically, the simulations are performed based on an emission inventory for the year of 2006. It is well known that China has made great effort in emission control in the past decade. For instance, $SO_2$ has been dropping sharply since 2006, and the $SO_2$ emission are estimated to be drop by over 70% in NCP in past 5 years due to the implementation of the toughest-ever clean air policy in China (Zhang et al., 2019). Is the emission inventory for the year of 2006 can represent the current emission scenario since the emissions underwent dramatic changes in both magnitude and spatial distribution in recent years?

Besides of photochemistry and heterogeneous chemistry, chemical production in cloud water is also an important contributor to secondary aerosol like sulfate and SOA. Why not track it in the SA calculation?

The descriptions on the model modifications need more detailed information and supporting references. 1. The yield value is vital for the simulating SOA but most references cited in this work is too outdated. Please specify the yield values from different VOCs to S/IVOC used in this simulation. 2. Line148-150, how the heterogenous oxidation of $SO_2$ in the aerosol water are parameterized? The aerosol water is not an ideal solution and thus all the classic reaction rate is not applicable here, and how the effect of ionic strength and aerosol water acidity that would significantly influence mass transfer are considered. 3. As mentioned, ISORRPIA is calculating aerosol thermodynamical equilibrium. How does the model attribute the chemical production from different sources since they are interacting with each other. The authors' writing style makes it quite hard to follow or repeat.

Another, the discussion of the results is a little descriptive, and more in-depth analysis and political implications are suggested here. For instance, is there any difference in source attributions at different altitude, and why? To control the air pollution in a more

cost-effective way, which kinds of emission sectors, like residential combustion and transportation, should be given priority over any other.

This work aims to discuss the contribution of local emissions and trans-boundary transport in NCP. Recent studies have demonstrated that the aerosol from cross-regional transport could exert substantial impacts on local meteorological condition in North China Plain, thereby deteriorating the PM2.5 pollution in this region. Such interaction has been also identified to be an important process in trans-boundary pollution (Huang et al., 2020). Can this source-oriented model resolve such kind process and quantify the relative contribution.

References:

Hu, J. L., Wu, L., Zheng, B., Zhang, Q., He, K. B., Chang, Q., Li, X. H., Yang, F. M., Ying, Q., and Zhang, H. L.: Source contributions and regional transport of primary particulate matter in China, Environ. Pollut., 207, 31-42, doi: 10.1016/j.envpol.2015.08.037, 2015.

Huang, X. et al., Amplified transboundary transport of haze by aerosol-boundary layer interaction in China, Nature Geoscience, doi:10.1038/s41561-020-0583-4, 2020.

Zhang, Q et al., Drivers of improved PM2.5 air quality in China from 2013 to 2017,Proceedings of the National Academy of Sciences Dec 2019, 116 (49) 24463-24469; DOI: 10.1073/pnas.1907956116

---

## Referee Comment (RC2) · Anonymous Referee #1 · 18 Aug 2020

Using a modified version of the WRF-Chem model, this paper utilizes a source apportionment method to examine particulate matter characteristics during a wintertime pollution event in China. The authors first compare the model output to chemistry and aerosol observations from ground-based sites and conclude that the model generally performs well. Then, the authors aim to explore the relative contributions of local and non-local emissions on air quality in various regions of China – a topic that is very important for emission regulations. Their findings suggest that for Beijing and Tianjin, local emissions tend to dominate when the air quality is excellent and good; however, the impact of non-local emissions becomes more pronounced as air quality decreases.

[Figure]

I think that the results stemming from this work are interesting and worthy of publication. In general, the paper is well written and the authors do a satisfactory job explaining their findings. However, I do have a few comments regarding several topics on which the authors could further elaborate. Overall, I recommend that the paper be accepted for publication once the authors address my comments. My major and minor concerns are described below, and my grammatical recommendations are provided in an associated PDF document.

Major/general comments: 1. Pollution event meteorology: In the context of this study, the transport of pollution is strongly dependent upon regional meteorology (e.g., advection of particulate matter by the mean wind). However, the authors do not put nearly enough emphasis on this topic. How do the large-scale meteorological conditions evolve over the course of the month? Surely, there was some variability; even just looking at Fig. 5a, one can hypothesize that there is some synoptic-scale influence. Please add a figure showing this evolution, perhaps near-surface pressure, winds, and temperatures at various snapshots during the event that correspond to the peaks and valleys in Fig. 5a. Moreover, the only discussion of wind flow is surrounding Fig. 6 and some other brief sections in the text. In Fig. 6, it appears as though the authors plot mean wind arrows for the pollution event. How do you calculate average wind direction during the time period? Do you think that it is valid to show a planview of average winds over a month-long period? Many of the regions show calm winds, but there is likely much variability over the course of the entire event. Showing a time series here of observed and/or modeled winds should help clarify. Additionally, regarding L210-212: Do you hypothesize that this is going on here? Are you able to use the surface measurements to determine if the modeled wind field is a major issue for this particular case? Again, perhaps a time series of wind speed/direction would help. Regarding L272-275: Did this occur during the case study examined here? The wind arrows in Fig. 6 suggest not, but it is difficult to tell since they are averages.

2. Figure 5: Because there are so many sites, it would be nice to see the spread

among sites. Is the model doing well at all of the sites? Or are many sites under- and over-predicted to "average out" and make it look like they are doing well? Does the model do well in one region over another? I suggest that you add a figure with panels showing scatter plots of these chemical species that compare observations and model for all sites and color by region.

3. Source apportionment uncertainty: Are you able to quantify the uncertainty in your source apportionment calculations? For instance, in Tables 2-5 and Figs. 9 and 10, can you add some information that helps understand the error in your estimates? For instance, add ranges in the tables and error bars on the bar plot figures.

Minor/specific comments: 1. L97-100: At the end of section 1, please provide a brief description of what you will present in the following sections.

2. Figure 1: I do not see any "blue circles", maybe you mean to say "circles".

3. Figure 1: What is the total number of sites considered in the analysis? This would be important to know also for Fig. 5.

4. L199-200: Do you allow for model spin-up? Table 1 says that the model start time is 05 December 2015, but you show results starting on this day.

5. Figures 6 and 8: What about the diurnal variability in the spatial distributions?

6. L225: Do you have evidence of cloud coverage during this event?

7. L232-233: Why do you choose to focus on the NCNST site?

Grammatical/wording recommendations: Please see the attached PDF.

Please also note the supplement to this comment:
https://acp.copernicus.org/preprints/acp-2020-597/acp-2020-597-RC2-supplement.pdf

[Figure]

**Supplement:**

[revised manuscript text omitted]

---

## Author Comment (AC1) · 17 Nov 2020

**Reply to Anonymous Referee #1**

We thank the reviewer for the careful reading of the manuscript and helpful comments. We have revised the manuscript following the suggestion, as described below.

Source attribution of air pollution is of great importance in emission control policy making. This work developed a source-oriented method in WRF-Chem regional model and applied it in the source appointment of fine particle pollution in the highly-pollution North China Plain region. Based on a one-month simulation using this source-oriented version of the WRF-Chem model, the authors indicated different contributions from local and non-local emissions for Beijing, Tianjin and other provinces and highlight the cooperation among provinces. Overall, this work is well structured but still needs more clarification and some in-depth analysis. Here are some issues that are suggested to be addressed for further improving this work.

**1 Comment**: Some detailed configurations of model need to be clarified and updated. The definition of source regions needs to be provided in the method. There is no information or figure of the model domain and designation of source regions. See Fig.1 in Hu et al. (2015). Also, technically, the simulations are performed based on an emission inventory for the year of 2006. It is well known that China has made great effort in emission control in the past decade. For instance, $SO_2$ has been dropping sharply since 2006, and the $SO_2$ emission are estimated to be drop by over 70% in NCP in past 5 years due to the implementation of the toughest-ever clean air policy in China (Zhang et al., 2019). Is the emission inventory for the year of 2006 can represent the current emission scenario since the emissions underwent dramatic changes in both magnitude and spatial distribution in recent years?

**Response:** We have added figure S1 to show designation of the source region and clarified in Section 3.2: "*We have marked the emitted precursors in six provinces, including Beijing, Tianjin, Hebei, Henan, Shandong, and Shanxi in simulations of the source-oriented WRF-Chem model (Figure S1).*".

We have clarified in Section 2.1: "*It is worth noting that the emission inventory used in this study is developed by Zhang et al. (2009) and Li et al. (2017) with the base year of 2012. Considering that the great changes in emission inventory due to implementation of the toughest-ever clean air policy in China (Zhang et al., 2019), the emission inventory has been adjusted according to the trends from 2012 to 2015 proposed by Zheng et al. (2018).*".

**2 Comment:** Besides of photochemistry and heterogeneous chemistry, chemical production in cloud water is also an important contributor to secondary aerosol like sulfate and SOA. Why not track it in the SA calculation?

**Response:** We have explained in Section 2.2: "*It is worth noting that, although it is lack of precipitation during the simulated episode, the SA of sulfate formed in cloud water is also considered. The $SO_2$ in cloud water is oxidized mainly by $H_2O_2$, $O_3$, $NO_2$, formic acid, and $O_2$ catalyzed by $Fe^{3+}$ and $Mn^{2+}$.*". We have considered the heterogeneous SOA formation from glyoxal and methyglyoxal on aerosol or cloud droplet surfaces with a reactive uptake coefficient of $3.7 \times 10^{-3}$.

**3 Comment:** The descriptions on the model modifications need more detailed information and supporting references. 1. The yield value is vital for the simulating SOA but most references cited in this work is too outdated. Please specify the yield values from different VOCs to S/IVOC used in this simulation. 2. Line148-150, how the heterogenous oxidation of $SO_2$ in the aerosol water are parameterized? The aerosol water is not an ideal solution and thus all the classic reaction rate is not applicable here, and how the effect of ionic strength and aerosol water acidity that would significantly influence mass transfer are considered. 3. As mentioned, ISORRPIA is calculating aerosol thermodynamical equilibrium. How does the model attribute the chemical production from different sources since they are interacting with each other? The authors' writing style makes it quite hard to follow or repeat.

**Response:**

1.  We have clarified in Section 2.2: "*The SOA yield from VOCs is NOx dependent (Li et al., 2011a). The high-NOx and low-NOx yields are listed in the Table S1 and parameters used to treat partitioning of POA emissions are listed in Table S2.*".

2. We have clarified in Section 2.2:"*In this study, a $SO_2$ heterogeneous reaction parameterization associated with aerosol water is used, in which the $SO_2$ oxidation in aerosol water by $O_2$ catalyzed by $Fe^{3+}$ is limited by mass transfer resistances in the gas-phase and the gas-particle interface. Considering the effect of ionic strength and aerosol water acidity, the sulfate heterogeneous formation from $SO_2$ is therefore parameterized as a first-order irreversible uptake by aerosols, with a reactive uptake coefficient of $0.5 \times 10^{-4}$, assuming that there is enough alkalinity to maintain the high iron-catalyzed reaction rate (Li et al., 2017). The detailed description of the parameterization of the heterogeneous oxidation of $SO_2$ involving aerosol water can be seen in Supplement.*". We have provided the detailed description in Section S1.

3. We have explained in Section 2.2: "*Therefore, as a bulk method, the ISORROPIA cannot be applied to distribute the gas and aerosol phase for the inorganic aerosol from each source separately because of the interaction among various sources.* ", and "*The SA for nitrate and ammonium aerosols follows the mass conversion of $N(+VI)$ and $N(-III)$ from each source, respectively, when the total ammonia and nitrate are distributed between the gas and aerosol phases by the ISORROPIA after one time step integration, as shown in Figure 3.*"

**4 Comment:** Another, the discussion of the results is a little descriptive, and more in-depth analysis and political implications are suggested here. For instance, is there any difference in source attributions at different altitude, and why? To control the air pollution in a more cost-effective way, which kinds of emission sectors, like residential combustion and transportation, should be given priority over any other.

**Response:**

1. We have added in Section 3.2: "*Figure S20 also provides the vertical profiles of the average $PM_{2.5}$ contribution from local and non-local emissions in Beijing, Tianjin, Hebei, Henan, Shandong, and Shanxi during the episode. Generally, the $PM_{2.5}$ contribution of local emissions in the six provinces in the NCP declines rapidly with altitude due to the efficient advection in the upper PBL. The local contribution decreases to less than 20% in the upper PBL in Beijing and Tianjin and is generally more than 25% in the other four provinces. In Shandong, the $PM_{2.5}$ concentration is mainly dominated by local emissions*

*in the lower PBL, but the local contribution presents a significant decreasing trend in the upper PBL.*"

2. We have added in Summary and conclusions: "*In this study, the source-oriented WRF-Chem model is also used to mark the precursors emitted from residential, transportation, industry, power, and agriculture sectors, respectively, to evaluate the contribution of anthropogenic emissions to the $PM_{2.5}$ concentration in the NCP. The average contribution of residential emissions to the $PM_{2.5}$ level is the most significant, with a maximum exceeding 100 $\mu g$ $m^{-3}$ during the study episode (Figure S21). In addition, the contribution of industry emissions to $PM_{2.5}$ concentration in the NCP also varies from 10 from 100 $\mu g$ $m^{-3}$ during the study episode. Therefore, more attention should be paid to residential and industry sectors to control the air pollution in a more cost-effective way.*" and "*The contribution of residential and industry emissions to the $PM_{2.5}$ concentration in Hebei, Shandong, and Henan is the most obvious (Figure S21). Therefore, efficient emission mitigations of air pollutants in the three provinces need to be carried out continuously to lower PM levels.*".

**5 Comment:** This work aims to discuss the contribution of local emissions and trans-boundary transport in NCP. Recent studies have demonstrated that the aerosol from cross-regional transport could exert substantial impacts on local meteorological condition in North China Plain, thereby deteriorating the $PM_{2.5}$ pollution in this region. Such interaction has been also identified to be an important process in trans-boundary pollution (Huang et al., 2020). Can this source-oriented model resolve such kind process and quantify the relative contribution.

**Response:** We have clarified in Summary and conclusions: "*The developed source-oriented model is mainly used in this study to quantitatively evaluate the local and non-local contributions to the PM pollution in the NCP. A recent study (Huang et al., 2020) has demonstrated that, absorption aerosols contributed by cross-regional transport from the Yangtze River Delta (YRD) to the upper PBL in the NCP induce the aerosol-PBL interaction and further lead to the suppressed PBL height, notable reduction of temperature and a substantial enhancement of relative humidity, favoring secondary aerosol production and aggravation of air pollution in the NCP. In this study, a sensitivity study without BC transported from the south of 32°N is conducted to analyze the contribution of the effect of*

*cross-regional transport of air pollutants on local meteorological conditions during the selected simulated episode. The temperature and PBL height decrease in the NCP caused by the BC transported from the south are not significant, with a maximum of 0.04 ℃ and 1.6%, and the increase of relative humidity just varies from -0.2% to 0.1% (Figure S22). Therefore, the aerosol-PBL interaction induced by the trans-boundary transport of absorption aerosols can not be observed in this study. In the future, more typical air pollution episodes need to be simulated to quantify the impact of regional transport of absorption aerosols on meteorological conditions.*"

**References:**

Hu, J. L., Wu, L., Zheng, B., Zhang, Q., He, K. B., Chang, Q., Li, X. H., Yang, F. M., Ying, Q., and Zhang, H. L.: Source contributions and regional transport of primary particulate matter in China, Environ. Pollut., 207, 31-42, doi: 10.1016/j.envpol.2015.08.037, 2015.

Huang, X., Ding, A., Wang, Z., Ding, K., and Fu, C.: Amplified transboundary transport of haze by aerosol–boundary layer interaction in China, Nat. Geosci., 13, 1-7, doi:10.1038/s41561-020-0583-4, 2020.

Zhang, Q., Zheng, Y. X., Tong, D., Shao, M., Wang, S. X., Zhang, Y. H., Xu, X. D., Wang, J. N., He, H., Liu, W. Q., Ding, Y. H., Lei, Y., Li, J. H., Wang, Z. F., Zhang, X. Y., Wang, Y. S., Cheng, J., Liu, Y., Shi, Q. R., Yan, L., Geng, G. N., Hong, C. P., Li, M., Liu, F., Zheng, B., Cao, J. J., Ding, A. J., Gao, J., Fu, Q. Y., Huo, J. T., Liu, B. X., Liu, Z. R., Yang, F. M., He, K. B., and Hao, J. M.: Drivers of improved $PM_{2.5}$ air quality in China from 2013 to 2017, P. Natl. Acad. Sci. USA., 116, 24463-24469, 10.1073/pnas.1907956116, 2019.

---

## Author Comment (AC2) · 17 Nov 2020

**Reply to Anonymous Referee #2**

We thank the reviewer for the careful reading of the manuscript and helpful comments. We have revised the manuscript following the suggestion, as described below.

Using a modified version of the WRF-Chem model, this paper utilizes a source apportionment method to examine particulate matter characteristics during a wintertime pollution event in China. The authors first compare the model output to chemistry and aerosol observations from ground-based sites and conclude that the model generally performs well. Then, the authors aim to explore the relative contributions of local and non-local emissions on air quality in various regions of China - a topic that is very important for emission regulations. Their findings suggest that for Beijing and Tianjin, local emissions tend to dominate when the air quality is excellent and good; however, the impact of non-local emissions becomes more pronounced as air quality decreases. I think that the results stemming from this work are interesting and worthy of publication. In general, the paper is well written and the authors do a satisfactory job explaining their findings. However, I do have a few comments regarding several topics on which the authors could further elaborate. Overall, I recommend that the paper be accepted for publication once the authors address my comments. My major and minor concerns are described below, and my grammatical recommendations are provided in an associated PDF document.

**Major/general comments:**

**1 Comment:** Pollution event meteorology: In the context of this study, the transport of pollution is strongly dependent upon regional meteorology (e.g., advection of particulate matter by the mean wind). However, the authors do not put nearly enough emphasis on this topic. How do the large-scale meteorological conditions evolve over the course of the month? Surely, there was some variability; even just looking at Fig. 5a, one can hypothesize that there is some synoptic-scale influence. Please add a figure showing this evolution, perhaps

near-surface pressure, winds, and temperatures at various snapshots during the event that correspond to the peaks and valleys in Fig. 5a. Moreover, the only discussion of wind flow is surrounding Fig. 6 and some other brief sections in the text. In Fig. 6, it appears as though the authors plot mean wind arrows for the pollution event. How do you calculate average wind direction during the time period? Do you think that it is valid to show a planview of average winds over a month-long period? Many of the regions show calm winds, but there is likely much variability over the course of the entire event. Showing a time series here of observed and/or modeled winds should help clarify. Additionally, regarding L210-212: Do you hypothesize that this is going on here? Are you able to use the surface measurements to determine if the modeled wind field is a major issue for this particular case? Again, perhaps a time series of wind speed/direction would help. Regarding L272-275: Did this occur during the case study examined here? The wind arrows in Fig. 6 suggest not, but it is difficult to tell since they are averages.

**Response:** We have added the description of the meteorological data in Section 2.3: "*The meteorological parameters including surface pressure, temperature, wind speed and direction with a 3-hour interval are obtained from the website http://www.meteomanz.com, including the observation sites at Beijing, Tianjin, Shijiazhuang, Jinan, Zhengzhou, Hefei, and Nanjing (Figure S1). Furthermore, the reanalysis data from the European Centre for Medium-Range Weather Forecasts (ECMWF) are used to analyze the synoptic patterns during the study episode.*".

We have clarified in Section 3.1: "*Generally, the accumulation and trans-boundary transport of air pollutants is mainly dependent on regional meteorological conditions. Figure S2 shows the average geopotential heights at 500hPa and the mean sea level pressures with wind vectors during the study episode. During the simulated episode, the NCP is situated behind the trough at 500 hPa. The NCP is controlled by the high pressure system at the surface on a large scale due to the upper level trough, ranging from 1026 to 1030 hPa, and the prevailing wind over the NCP is weak or calm, which is unfavorable for dissipation of air pollutants. Figure S3 shows the diurnal profiles of observed and simulated near-surface pressure, temperature, wind speed, and wind direction averaged at monitoring sites in the NCP from 05 December 2015 to 04 January 2016. The WRF-Chem model performs well in reproducing the diurnal variability of near surface pressure, surface temperature (TSFC), wind speed, and wind direction, with IOAs of 0.63, 0.84, 0.75, and 0.54, respectively. During the study episode, the simulated and observed of near surface pressures are 1024.0hPa and 1028.5hPa,*

*indicating that a high pressure system controlling the NCP (Figure S2). The southerly wind prevails over the NCP during the study episode, with the simulated and observed wind direction of 180.6° and 175.1°. Moreover, the simulated and observed wind speed is approximately 2 m s$^{-1}$ over the NCP during the simulated episode. Therefore, the air pollutants are subject to being transported from south to north, and the weak or calm wind also appears in some regions, which is favorable for the accumulation of air pollutants. For example, from 16 to 24 December 2015, the wind speed in the NCP decreases and the wind direction turns to be southerly, facilitating accumulation of air pollutants, and meanwhile a serious PM pollution episode with high PM$_{2.5}$ concentrations occurs.".*

We have clarified the calculation of average of wind direction in Supplement Section S3: *"The wind direction simulated in this study is calculated using the U (the velocity toward east) and V (the velocity toward north) component at a specific grid point over the simulation domain and the average wind direction is calculated based on the average U and V.".*

**2 Comment:** Figure 5: Because there are so many sites, it would be nice to see the spread among sites. Is the model doing well at all of the sites? Or are many sites under- and over-predicted to "average out" and make it look like they are doing well? Does the model do well in one region over another? I suggest that you add a figure with panels showing scatter plots of these chemical species that compare observations and model for all sites and color by region.

**Response:** We have clarified in Section S2.1 *Air pollutants simulations in different cities in the NCP: "Considering that there are many monitoring sites in the NCP, scatter plots of observed and simulated PM$_{2.5}$, O$_3$, NO$_2$, SO$_2$ and CO concentrations for all sites in Beijing, Tianjin, Hebei, Henan, Shandong, Shanxi, Jiangsu, and Anhui from 05 December 2015 to 04 January 2016 have also been provided in Figures S4 to S8, respectively. Except Anhui, the correlation coefficients between observed and simulated PM$_{2.5}$ concentrations are generally larger than 0.70 (Figure S4). The model also performs well in simulating the O$_3$ concentration in the NCP, with correlation coefficients generally larger than 0.80 (Figure S5). The NO$_2$ concentration in the NCP is also simulated reasonably, with correlation coefficients generally ranging from 0.70 to 0.80 (Figure S6). Considering that the SO$_2$ is mainly emitted from point sources, which is more sensitive to meteorological conditions, the model has difficulties in simulating the SO$_2$ concentration, with correlation coefficients generally less*

*than 0.60 (Figure S7). In addition to Tianjin and Shanxi, the CO concentration is also reasonably reproduced, with correlation coefficient larger than 0.70 (Figure S8).".*

**3 Comment:** Source apportionment uncertainty: Are you able to quantify the uncertainty in your source apportionment calculations? For instance, in Tables 2-5 and Figs. 9 and 10, can you add some information that helps understand the error in your estimates? For instance, add ranges in the tables and error bars on the bar plot figures.

**Response:** We have added uncertainty in the tables and error bars on the bar plot figures in Tables 2-5 and Figs. 9 and 10.

**Minor/Specific comments:**

**1 Comment:** L97-100: At the end of section 1, please provide a brief description of what you will present in the following sections.

**Response:** We have added at the end of Section 1: "*The model and methodology are described in Section 2. The results and discussions are presented in Section 3, and summary and conclusions are given in Section 4.*".

**2 Comment:** Figure 1: I do not see any "blue circles", maybe you mean to say "circles".

**Response:** We have revised the figure caption of Figure 1"T*he circles represent centers of cities with ambient monitoring sites, and the size of blue circles denotes the number of ambient monitoring sites of cities.*" as "*The circles represent centers of cities with ambient monitoring sites, and the size of circles denotes the number of ambient monitoring sites of cities.*".

**3 Comment:** Figure 1: What is the total number of sites considered in the analysis? This would be important to know also for Fig. 5.

**Response:** We have added in the Section 2.3: "*The model performance in simulating PM₂.₅, O₃, NO₂, SO₂, and CO is validated using the hourly observations released by Ministry of Ecology and Environment of China (China MEP), with 389 observation sites in the NCP.*".

**4 Comment:** L199-200: Do you allow for model spin-up? Table 1 says that the model start time is 05 December 2015, but you show results starting on this day.

**Response:** Yes, we have allowed for model spin-up. The spin-up time is 4 days and 4 hours, and the simulation period starts from 05 December 2015. We have updated Table 1.

**5 Comment:** Figures 6 and 8: What about the diurnal variability in the spatial distributions?

**Response:** We have added the diurnal variability in the spatial distributions and clarified in Section 3.1: "*The diurnal variability in the spatial distribution of simulated and observed air pollutants is shown in Figures S9 to S12. The spatial patterns of air pollutants at different time are generally similar to those of the episode average. The PM₂.₅ pollution in the NCP is more severe during nighttime and early morning, especially at 08:00 and 20:00 BJT due to the rush hour.*" and Section 3.2: "*The diurnal variations in the spatial distribution of average PM₂.₅ contributions from the six provinces during the study episode are also shown in Figures S14 to S19. There is no significant difference among the spatial distribution of PM₂.₅ contributions from the six provinces at different time, but the higher PM₂.₅ contribution of emissions from the source region generally occurs at 08:00 and 20:00 BJT.*".

**6 Comment:** L225: Do you have evidence of cloud coverage during this event?

**Response:** We have clarified in Section S2.2 Cloud properties: "*Clouds are one of the most important factors affecting the solar radiation reaching the ground. The daily cloud fraction (CF) used in this study was retrieved from Terra- and Aqua- Moderate Resolution Imaging Spectroradiometer (MODIS) level 2 products. Figure S13 presents the scatter plot of the daily retrieved and simulated CF averaged in the NCP from 05 December 2015 to 31 December 2015. Generally, the simulated daily average CF correlates well with that retrieved, with a correlation coefficient of 0.69. The simulated average CF over the NCP during the episode is*

*52.8%, lower than the MODIS retrieved 78.4%. Numerical models still have difficulties in representing accurately clouds in terms of microphysical processes, cloud morphologies, occurrence and dissipation. In addition, many uncertainties also significantly impact CF retrievals, such as the satellite's view zenith angle, cloud microphysics assumptions, namely cloud phase, particle size and shape, et al. (An and Wang, 2015; Platnick et al., 2017; Zeng et al., 2012; Li et al., 2014). Therefore, it is still difficult to validate cloud simulations using the satellite cloud products.* ".

**7 Comment:** L232-233: Why do you choose to focus on the NCNST site?

**Response:** The hourly submicron sulfate, nitrate, ammonium, and organic aerosols are measured by the Aerodyne Aerosol Chemical Speciation Monitor (ACSM) at NCNST, which are used to validate the model performance.

Grammatical/wording recommendations: Please see the attached PDF.

Please also note the supplement to this comment:

https://acp.copernicus.org/preprints/acp-2020-597/acp-2020-597-RC2- supplement.pdf.

**Response:** Thanks. We have revised the manuscript according to the attached comment